# The Study of Protein–Cyclitol Interactions

**DOI:** 10.3390/ijms23062940

**Published:** 2022-03-09

**Authors:** Tetiana Dyrda-Terniuk, Mateusz Sugajski, Oleksandra Pryshchepa, Joanna Śliwiak, Magdalena Buszewska-Forajta, Paweł Pomastowski, Bogusław Buszewski

**Affiliations:** 1Department of Environmental Chemistry and Bioanalytics, Faculty of Chemistry, Nicolaus Copernicus University in Toruń, Gagarina 7, 87-100 Toruń, Poland; tania25dyrda@gmail.com (T.D.-T.); mateusz.sugajski@o2.pl (M.S.); pryshchepa.alexie@gmail.com (O.P.); bbusz@chem.umk.pl (B.B.); 2Interdisciplinary Centre of Modern Technologies, Nicolaus Copernicus University in Toruń, Wileńska 4, 87-100 Toruń, Poland; 3Institute of Bioorganic Chemistry, Polish Academy of Sciences, Noskowskiego 12/14, 61-704 Poznań, Poland; sliwiak@gmail.com; 4Institute of Veterinary Medicine, Nicolaus Copernicus University in Toruń, Gagarina 7, 87-100 Toruń, Poland; m.buszewska@umk.pl

**Keywords:** BSA, cyclitol, mechanism, interactions, bonding

## Abstract

Investigation of interactions between the target protein molecule and ligand allows for an understanding of the nature of the molecular recognition, functions, and biological activity of protein–ligand complexation. In the present work, non-specific interactions between a model protein (Bovine Serum Albumin) and four cyclitols were investigated. D-sorbitol and adonitol represent the group of linear-structure cyclitols, while shikimic acid and D-(***–***)-quinic acid have cyclic-structure molecules. Various analytical methods, including chromatographic analysis (HPLC-MS/MS), electrophoretic analysis (SDS-PAGE), spectroscopic analysis (spectrofluorimetry, Fourier transform infrared spectroscopy, and Raman spectroscopy), and isothermal titration calorimetry (ITC), were applied for the description of protein–cyclitol interactions. Additionally, computational calculations were performed to predict the possible binding places. Kinetic studies allowed us to clarify interaction mechanisms that may take place during BSA and cyclitol interaction. The results allow us, among other things, to evaluate the impact of the cyclitol’s structure on the character of its interactions with the protein.

## 1. Introduction

Cyclitols are known as plant secondary metabolites whose distribution in plants increases under stress conditions. Cyclitols mediate numerous essential processes in plant cells, e.g., signal transduction, osmoregulation, formation of the cell wall, and phosphate storage [1]. These phytochemicals can be successfully utilized in the pharmaceutical industry because of their broad spectrum of health-promoting properties, including anti-diabetic, anti-inflammatory, antioxidative, and even anticancer effects [2]. Furthermore, cyclitols were also identified as effective therapeutic agents in the treatment of cardiovascular diseases, polycystic ovary syndrome, and psoriasis [3,4]. The present study was focused on four cyclitols with different structures that allow for an evaluation of their biological activity and prospective pharmacological features. Thus, cyclitols from two groups were chosen: linear-structure molecules (D-sorbitol and adonitol) and cyclic-structure molecules (shikimic acid and D-(***–***)-quinic acid). Sorbitol (also known as D-glucitol) is commonly found in various fruits (e.g., apple, peach, and cherry). It is naturally photosynthesized in leaves by enzymatic reduction of glucose 6-phosphate [5]. On a large scale, sorbitol is produced by glucose catalytic hydrogenation. Sorbitol belongs to the class of nutritional sweeteners, which are commonly used in diabetic food [6]. It participates in the regulation of glucose metabolism provided by incomplete intestinal sugar absorption and an increase in muscle glucose uptake [7]. Adonitol (or ribitol) is a pentose alcohol that can be found in Gram-positive bacteria cell walls, where it forms a complex in association with teichoic acids [8]. Adonitol in its free form has also been found in plants of the *Adonis* and *Bupleurum* genus [9]. Adonitol supplementation in dairy products could have a beneficial influence on human gut microbiota as it has a protective effect on lactic acid bacteria [10]. Moreover, adonitol treatment improves skeletal muscle functions and reduces the fibrosis in cardiac muscle [11]. Shikimic acid (3,4,5-trihydroxy-1-cyclohexene-1-carboxylic acid) is considered to be an important precursor for the synthesis of pharmaceutical agents with antiviral, antioxidant, and anticancer effects [12]. *Illicium Verum* (or the Chinese star anise tree) was reported to be the main plant source of shikimic acid. It also can be enzymatically obtained from quinic acid or metabolically produced in *E. Coli* [13]. Shikimic acid could be applied as a preservative in food because of its effective antibacterial properties against pathogenic species [14]. D-(***–***)-quinic acid (1,3,4,5-tetrahydroxycyclo-hexane-L-carboxylic acid) is naturally present in plant sources, e.g., coffee beans, kiwifruit, prunes, and bilberry [15]. Various in vivo and in vitro studies have confirmed the broad range of biological activities of D-(***–***)-quinic acid. Cyclitol supplementation provides an enhancement of the antioxidant status and in DNA repair [16]. It was reported that D-(***–***)-quinic acid also is capable of inhibiting the replication of Hepatitis B virus [17]. 

Serum albumins (SAs) belong to the dominant group of serum proteins. SAs are initially produced by the parenchymal cells in the liver and then are mainly excreted to blood plasma. Nevertheless, trace amounts of SAs are also found in milk [18]. Bovine Serum Albumin (BSA) is a relatively large globular protein whose molecular weight ranges between 66 and 67 kDa [19]. BSA consists of a single protein chain composed of 583 amino acids, including 17 disulphide bridges and 1 free cysteine residue. The thermodynamically stable three-dimensional structure of BSA is maintained by a variety of non-covalent interactions between amino acid residues and solvent molecules. BSA is defined as a heart-shaped protein that consists of three homologous domains (I, II, and III). Each domain is divided into two subdomains (A and B). Two specific hydrophobic regions of BSA in the IIA and IIIA subdomains (also known as Sudlow’s sites) are responsible for small molecule binding [20]. BSA was reported to be the major target protein in blood for various exogenous and endogenous compounds, which facilitates their transport to and distribution in the cells [21]. Importantly, BSA also provides a high level of bioavailability for hydrophobic bioactive substances. Numerous studies have already described the role of molecular interactions of BSA with metal nanoparticles, fatty acids, vitamins, flavonoids, and pharmaceuticals [18,22,23,24]. Hence, BSA is considered to be one of the most commonly used model proteins for studying protein–ligand interactions [18].

Protein–ligand interactions play an essential role in cellular metabolism, controlling various physiological processes such as enzyme catalysis, signal transduction, and antigen–antibody interactions [25]. The biological activity of a macromolecule can be modulated after ligand association to a particular site on the protein [26]. The binding site (binding pocket) usually has a concave shape with a characteristic amino acid arrangement that occupies only a small part of the protein surface. Even a slight change in its position and composition can significantly influence the affinity and specificity for ligands. Determination of the character of the interaction between the target protein and the ligand is of great importance to the establishment of the relationship between the composition of the binding site and ligand specificity [27]. For example, aromatic compounds containing charged substituents, such as carboxylic groups (salicylic acid, benzoic acid), interact with BSA via hydrogen bonding, electrostatic interactions, and van der Waals forces. However, in the case of heterocyclic rigid molecules (coumarin, carbamazepine) hydrophobic interactions are preferred [28,29]. Investigation of the mechanisms of ligand binding to BSA and evaluation of the structural features of the formed complexes could constitute a prospective approach in drug design and lead to the discovery of new drug candidates. Cyclitols represent such candidates. Previous studies have confirmed their stabilization effect on the structure of globular proteins. It was reported that cyclitols promote the folding of the macromolecule as their interactions with the peptide backbone have an unfavorable character. Thus, the Gibbs free energy of protein denaturation tends to be increased in the presence of cyclitols [30,31]. Understanding protein–cyclitol interactions enables the characterization of the conformational stability of BSA as a potential ligand delivery system in a therapeutic approach. Moreover, it clarifies the importance of cyclitols to biological processes.

The present work aimed to study the mechanism of interactions between BSA and cyclitols with the use of complementary instrumental techniques, including high-performance liquid chromatography coupled with tandem mass spectrometry (HPLC-MS/MS), sodium dodecyl sulfate–polyacrylamide gel electrophoresis (SDS-PAGE), spectroscopic analysis (spectrofluorimetry, FTIR, and Raman spectroscopy), and isothermal titration calorimetry (ITC). Additionally, molecular docking studies were performed.

## 2. Results and Discussion

### 2.1. SDS-PAGE

SDS-PAGE allows for the observation of the molecular weight of native BSA and BSA–cyclitol complexes. The bands on the obtained electropherogram (Figure 1) are visible at about 65 kDa and 130 kDa. The results reveal that the analyzed protein was present in the monomeric as well as in the dimeric form. It is well known that commercial preparations of BSA frequently contain protein oligomers [19]. In our study, the same bands also appeared for protein–cyclitol samples in comparison with the control. Thus, additional polymerization and aggregation processes were not induced by cyclitols. Based on the results, we can conclude that the tested cyclitols did not have a destabilization and degradation effect on the protein’s native structure [32,33].

### 2.2. Kinetic Modeling of BSA–Cyclitol Interactions

A kinetic study was used to describe the interactions between BSA and cyclitols with linear and cyclic structures. The results are presented in Figure 2 as the zero-order kinetic model. The model shows the change in the pH of the protein solution after cyclitol addition per unit time. The measured values of pH for control solutions of cyclitols were 5.56 (adonitol), 5.49 (D-sorbitol), 4.01 (shikimic acid), and 3.79 (D-(***–***)-quinic acid). Protein represents a complex biocolloidal system in which the electrical charge is strongly dependent on environmental conditions, especially pH [34]. The initial pH of the protein solution was 7.14; thus, the BSA molecule was negatively charged. The pH in the BSA control solution did not change during the entire experiment. In turn, a decrease in pH after cyclitol addition was observed in all other samples. This could be due to preferential interactions between cyclitol and water molecules and acidic side chains of aspartic acid (pKa = 3.9), glutamic acid (pKa = 4.3), and terminal α-carboxylic groups (pKa = 1.8–2.9) of the protein [35,36]. Cyclitols could mediate the preferential binding of the solvent (water) onto the BSA particle [37].

Autoionization of water molecules promotes an increase in hydroxyl groups’ availability on the protein surface, while hydronium ions participate in the neutralization of deprotonated carboxyl groups. The following hydrolysis of carboxyl groups led to a sharp pH decrease. The sorption process of adonitol (Figure 2A), D-(***–***)-quinic acid (Figure 2B), and D-sorbitol (Figure 2C) was non-homologous and occurred in two steps. For linear cyclitols, firstly, it was rapid sorption (Step I), and then gradual sorption (Step II) [38]. The rapid sorption stage took about 30 min. The calculated values of the zero-order rate constants of each step were very close for both cyclitols (Step I: 3.1 × 10^−2^ min^−1^ and 3.0 × 10^−2^ min^−1^; Step II: 5.7 × 10^−5^ min^−1^ and 6.4 × 10^−5^ min^−1^ for D-sorbitol and adonitol, respectively). Based on the obtained results, it was considered that the mechanism of interactions between these cyclitols and BSA had the same character. The more visible decrease in the pH of the BSA–shikimic acid and BSA–D-(***–***)-quinic acid systems is related to the presence of an additional carboxylic group in the cyclitol structure that could participate in the hydrolysis process. In the case of D-(***–***)-quinic acid, we observed the fastest decrease in pH (k_0_ = 1.92 min^−1^) and then a slight increase in pH. The reason for such changes could be the equilibrium of hydrogen bonding and electrostatic interactions between BSA and D-(***–***)-quinic acid that occurs with the participation of the hydroxyl and carboxylic groups bound to the same carbon atom of the cyclitol. BSA–shikimic acid complex (Figure 2D) formation was a one-step process of rapid sorption followed by immediate establishment of equilibrium. Further possible interactions may be blocked by the enhancement of repulsion forces between the carboxylic group of the cyclitol and negatively charged groups of BSA.

The Weber–Morris intraparticle diffusion model was applied to describe the mechanism of protein–cyclitol interactions (Figure 3). The Weber–Morris model displayed two steps of sorption in the case of D-sorbitol and adonitol (Figure 3A,B). The first step represents the binding of the cyclitol onto the external surface of the protein, boundary layer diffusion effects, and a reduction in the amount of cyclitol in the solution [39]. The speed of the process in the second step becomes limited, thus confirming the intraparticle diffusion process that is accompanied by an increase in the number of hydroxyl groups inside the protein structure [38]. The binding process of shikimic acid and D-(***–***)-quinic acid (Figure 3C,D) according to this model revealed only one step of the cyclitol binding process, which was attributed to sorption on the external BSA surface. This means that interactions of these cyclitols with BSA induce an increase in hydroxyl groups’ availability exclusively on the protein surface. The intraparticle diffusion rate constant K_ip_ and the A constant, expressed as the slope and *y*-axis intercept, respectively, of each plot, were calculated using the LINEST function. In the case of linear cyclitols, the calculated values of the intraparticle diffusion rate constant K_ip_ were higher than for molecules with a cyclic structure. The absence of an intraparticle diffusion stage during the sorption of shikimic acid and D-(***–***)-quinic acid could explain their relatively low K_ip_ values [40]. The obtained values of the A constant indicate the thickness of the boundary diffusion layer, which is correlated to the amount of bounded cyclitol on the external protein surface in the initial stage [34]. Based on these results, we can conclude that shikimic acid and D-(***–***)-quinic acid, due to their rigid structure and steric exclusion, are preferentially bounded onto the external protein surface. However, cyclitols with a linear structure are partially adsorbed inside the BSA structure [41].

The calculated Gibbs free energy change (ΔG) of cyclitol sorption by BSA obtained with the utilization of equation (9) is presented in Table 1. Negative values of ΔG indicate that cyclitol binding to the protein occurred spontaneously [42]. Protein targeting by small molecules is regulated mainly by non-covalent interactions. The calculated Gibbs free energy change for protein–cyclitol binding ranged from −27 kJ/mol to −21 kJ/mol and could be attributed to relatively weak interactions, such as hydrogen bonding, van der Waals forces, and hydrophobic, steric, and solvent-mediated interactions [43,44]. Thus, despite steric exclusion, cyclic-structure molecules are more strongly bound to the protein surface than linear cyclitols [36].

### 2.3. HPLC-MS/MS Analysis 

#### 2.3.1. Analysis of Protein–Cyclitol Interactions

High-performance liquid chromatography coupled with tandem mass spectrometry (HPLC-MS/MS) was applied for quantitative analysis of BSA–cyclitol interactions. A column filled with BSA protein particles as a stationary phase and a series of modified mobile phases, including water, water with formic acid, water with isopropanol, and a mixture of water, formic acid, and isopropanol, were used for the determination of retention parameters and identification of pseudomolecular ions (Table 2). All samples were monitored in the Selected Ion Monitoring (SIM) mode.

Adonitol was the only cyclitol identified in the positive ionization mode (1), while the rest of the cyclitols were analyzed in the negative ionization mode (2). The application of several mobile phases also allowed us to observe sodium adducts of the monomer and the dimer (only for adonitol) and dimers (for D-sorbitol and D-(–)-quinic acid).
[M] + H^+^ → [M − H]^+^,(1)
[M] → [M]^−^ + H^+^,(2)

Water and mixtures of water and isopropanol provided the identification of only the cyclitols with a linear structure. Additionally, the isomer of D-sorbitol was registered. The modification of the mobile phase with formic acid facilitated the ionization process of cyclitols and all phases containing FA enabled the identification of all analytes [45]. Thus, based on these data, the water + 0.1% *v/v* FA mobile phase was selected as the most optimal.

Figure 4 shows chromatograms of adonitol, D-sorbitol, shikimic acid, and D-(***–***)-quinic acid. The retention times of shikimic acid (t_R_ = 5.099) and D-(***–***)-quinic acid (t_R_ = 10.273) were the highest. This may be a result of their cyclic structure and steric interactions [44]. The contribution of electrostatic interactions between charged groups of BSA and carboxylic groups of shikimic acid (pKa = 4.1) and D-(***–***)-quinic acid (pKa = 3.5) was negligible, as the measured pH of the mobile phase was 2.61. However, shikimic acid contains a double bond in its structure that can be involved in the π–π stacking interactions with aromatic amino acids [43]. The presence of an additional hydroxyl group in the D-(***–***)-quinic acid structure, which neighbors the carboxyl group, can significantly affect the strength of interactions with BSA. Such a high affinity for BSA can also be correlated to the highest value of the rate constant in the initial step of the cyclitol sorption kinetic. In the case of D-sorbitol (t_R_ = 1.550; t_R_ = 2.621) and adonitol (t_R_ = 2.626), the retention times were the lowest. Thus, for the cyclitols with a linear structure much weaker interactions (hydrogen bonding, van der Waals forces) with BSA are characteristic. Moreover, this was confirmed by the calculated values of the Gibbs free energy change [46]. The protein–cyclitol sorption process is accompanied by solvent-mediated interactions, such as preferential exclusion, which is characteristic of polyols. The protein minimizes the surface in contact with small polar molecules because these interactions are thermodynamically unfavorable. Therefore, the free energy of protein unfolding significantly increases after interactions with cyclitols and the macromolecule tends to obtain a more compact structure [33,47].

#### 2.3.2. Indication of Unbounded Cyclitols

To calculate the amount of unbounded cyclitol in the reaction mixture, the calibration was performed without a column with the following mobile phase composition: 0.1% (*v*/*v*) formic acid–water (A) and acetonitrile (B) (75:25 (*v*/*v*). The amount of each unbounded cyclitol was calculated using calibration curves, which are presented in Table 3. The obtained calibration curve parameters have good linearity, and the R^2^ coefficients range between 0.9960 and 0.9985. Unfortunately, it was not possible to obtain a linear dependence for adonitol under the established conditions. The percentage of bounded cyclitol with BSA was calculated as the difference in the initial cyclitol concentration and its unbounded fraction. The number of bounded cyclitol molecules was also determined. The obtained results are shown in Table 3.

It was found that at a constant concentration of BSA, as the D-sorbitol concentration increased the percentage of bounded cyclitol firstly decreased, then increased, and finally decreased. One molecule of D-sorbitol is bound to BSA at lower cyclitol concentrations; however, at a higher molar ratio the protein binding effectiveness increased to two molecules of D-sorbitol. In the case of shikimic acid, the percentage of the bounded fraction with the protein decreased at higher cyclitol concentrations. Two molecules of shikimic acid are bound to BSA at low initial concentrations of cyclitol. The equilibrium of BSA–cyclitol complex formation was set at three molecules of the ligand. The sorption effectiveness of D-(***–***)-quinic acid was negligible at small concentrations of the ligand. However, it significantly increased as the initial concentration of cyclitol increased. One BSA particle was able to bind about four molecules of D-(***–***)-quinic acid. The protein was not fully saturated in this range of cyclitol concentrations and equilibrium was not established. This may be related to the colloidal character of the macromolecule. Cyclitols influence the thickness of the electric double layer of the BSA; thus, more of the protein surface becomes available for ligand binding [48].

### 2.4. Fluorescence Spectroscopy

Conformational changes in a protein’s structure can be monitored by the application of spectroscopic analysis. Spectrofluorimetry is an excellent technique that allows for analysis of the influence of ligands on the protein’s intrinsic fluorescence. Aromatic amino acids, including tryptophan, tyrosine, and phenylalanine, constitute the major protein fluorophores. BSA’s polypeptide chain contains 49 aromatic amino acid residues, including 2 tryptophans (Trp), 20 tyrosines (Tyr), and 27 phenylalanines (Phe). The absorptivity and the quantum yield of Phe are relatively low compared with tryptophan and tyrosine [49]. Hence, the contribution of Phe to protein fluorescence is not substantial. The obtained three-dimensional spectrum demonstrates the spectral features of native BSA and BSA–cyclitol complexes (Figure 5). BSA and its complexes undergo excitation at 225 nm and 280 nm, while the emission wavelength of both peaks was recorded at 340 nm. Peak 1 (λ_Ex_ = 225 nm and λ_Em_ = 340 nm) and Peak 2 (λ_Ex_ = 280 nm and λ_Em_ = 340 nm) display the local environment of aromatic amino acid side chains [50]. UV-VIS spectra of BSA and the cyclitols and 3D-fluorescence spectra of the cyclitols and solvent are shown in the Appendix A, respectively). Shikimic acid and D-(***–***)-quinic acid had a high level of absorption at 225 nm; thus, the fluorescence quenching of Peak 1 may have been burdened by the inner-filter effect [51]. Nevertheless, the cyclitols exhibited no light absorption at 280 nm. Hence, Peak 2 was chosen for more accurate monitoring of the conformational changes in BSA. The slight emission of cyclitol solutions (under 100 units) at 340 nm could be assigned to the fluorescence of the negative control (a cuvette with water). No band shifting was observed for the BSA–cyclitol samples, indicating that no visible changes in polarity occurred in the aromatic amino acid microenvironment [51]. The sharp decrease in the fluorescence intensity (quenching) of the BSA–cyclitol complexes in comparison with free BSA proves that protein–ligand interactions occurred and indicates that cyclitols are localized in close proximity to tryptophan and tyrosine residues. Still, these small molecules do not induce significant conformational changes in proteins [22]. The more effective fluorescence quenching of BSA in the presence of shikimic acid and D-(***–***)-quinic acid confirms the much stronger binding affinity of these cyclitols to the protein than D-sorbitol and adonitol [45].

### 2.5. FTIR Analysis

FTIR spectra of the analyzed samples in the range of 4000–950 cm^−1^ are presented in the Appendix A. FTIR spectra of native BSA, the cyclitols, and their complexes (Figure 6) show variations in the characteristic bands of active functional groups of the protein involved in the interactions in the fingerprint region. A detailed list of the captioned wavenumbers in Figure 6 is presented in the Appendix A. Spectral shifting of amide bands allows for the determination of the influence of cyclitols on the secondary structure composition of, conformational changes in, and the stability of a protein. The monitored changes in the BSA–cyclitol complex spectra may be successfully utilized for a comparison analysis with molecular docking results. As the experiment was not performed quantitively, the peak intensities of BSA and BSA–cyclitol complexes could not be compared. Cyclitols induced a peak shift in the amide A and amide I bands to higher frequencies, while the amide II band shifted to lower frequencies. Signals at 1490 and 1448 cm^−1^ indicated the new intermolecular interactions of the cyclitol with amino acid side chains. New intensive bands appeared in the range of 1200–1000 cm^−1^ (characteristic of C–O and O–H functional groups), which allows for confirmation of the cyclitol’s sorption onto the protein in the solution.

In Appendix A, it can be seen that the amide A band of the BSA control (3300 cm^−1^), which is responsible for N–H stretching vibrations, is sharper compared with the BSA–cyclitol samples. The peaks of the complexes become much wider in this range, which is a result of the imposition of the additional signals originating from hydroxyl groups of cyclitols. The shift in the amide A band indicates that the cyclitol’s interaction with the BSA polypeptide backbone has an influence on the strength of intramolecular hydrogen bonds [52]. The peaks in the 3000–2850 cm^−1^ range originate from C–H group vibrations. The peaks of the control sample at 2957 cm^−1^ can be assigned to asymmetric stretching vibrations of CH_3_. The asymmetric and symmetric stretching vibrations of CH_2_ (at 2918 and 2850 cm^−1^, respectively) originate from D-sorbitol and adonitol samples. However, the other two bands of the BSA–cyclitol samples are responsible for the symmetric stretching vibrations of CH_2_ (2934 and 2936 cm^−1^) and CH_3_ (2876 and 2878 cm^−1^) [53,54].

The fingerprint region (Figure 6) allows for a more detailed analysis of conformational changes in the BSA structure and a comparison of the amide I–III bands. The amide I band (1700–1600 cm^−1^) is mainly attributed to C=O stretching vibrations (80%) and C–N stretching vibrations. The position of the amide I band is very sensitive for determining the type of protein secondary structure [55]. The peaks observed in the range of 1660–1650 cm^−1^ are characteristic of an α-helix [56]. In our case, the addition of cyclitols led to a shift in this band toward higher frequencies from 1654 cm^−1^ to 1656 cm^−1^ and 1658 cm^−1^ (in the case of D-(–)-quinic acid), indicating cyclitol interactions with the polypeptide backbone [57]. This may also be a result of the less polar local environment of the protein as BSA forms much weaker hydrogen bonds with cyclitols compared with water [58]. The protein side chains, including aromatic, basic, and acidic amino acid residues, also contribute to adsorption in this region. The region is also sensitive for the monitoring of the protonation and deprotonation of the carboxylic groups of aspartic and glutamic acids, which participate in the sorption process of cyclitols. Protein–cyclitol interactions mediated by solvent provide a more hydrophobic environment for BSA and induce a shift in the peak towards higher frequencies [59,60,61].

The amide II band (1600–1500 cm^−1^) includes signals that originate from protein backbone in-plane N–H vibrations and C–N stretching vibrations. The region of registered signals could be assigned to an α-helical structure [62]. In the case of BSA–cyclitol samples (except for BSA/D-(–)-quinic acid), this band shifted to lower wavenumbers compared with native BSA (from 1548 to 1546 cm^−1^), indicating the formation of new hydrogen bonds between N–H groups of the protein and cyclitol molecules [63,64,65].

The weak band characteristic of stretching C–C ring vibrations of phenylalanine appears for BSA and BSA/D-sorbitol at 1492 and 1490 cm^−1^, respectively [52]. The bands at 1448 cm^−1^ (bending CH_3_ and stretching C–N vibrations of His^-^) indicate that D-sorbitol, adonitol, and D-(–)-quinic acid are involved in interactions with histidine [39]. However, the signal at 1454 cm^−1^ for the BSA/shikimic acid sample originates from CH_2_ scissor vibrations [53]. The signals at 1406, 1402, 1400, and 1392 cm^−1^ come from symmetric stretching vibrations of the deprotonated carboxyl group COO^-^ of glutamate and aspartate side chains. For BSA complexes with cyclitols, these bands shifted to higher wavenumbers, indicating a more hydrophobic protein environment [59]. These results confirm the mechanism of BSA–cyclitol interactions from kinetic studies.

The weak signal at 1356 cm^−1^ (bending C–H vibrations of tryptophan) of BSA was overlapped by the amide III band [59]. The range of 1350–1200 cm^−1^ is characteristic of the amide III band, which includes mainly stretching C–N and in-plane N–H bending vibrations. This region also displays the changes occurring in the protein secondary structure [66]. The band appears at 1294 cm^−1^ for the BSA control, which is responsible for the α-helix structure, while the peaks at 1272 and 1238 cm^−1^ can be assigned to the β-turn and β-sheet, respectively. New bands appear at 1312, 1304, and 1302 cm^−1^ (α-helix) and at 1246 and 1242 cm^−1^ (β-sheet) in the presence of cyclitols [66,67].

Weak signals were observed at 1122 cm^−1^ (stretching C–O vibrations of aspartic and glutamic acids), 1088 cm^−1^ (stretching C–N vibrations coupled with bending C–H vibrations of histidine), 1062 cm^−1^ (stretching C–O vibrations of threonine), 1000 cm^−1^ (symmetric stretching C–C ring vibrations of phenylalanine), and 972 cm^−1^ (stretching C–O vibrations of serine) for the protein sample [59,67,68]. This region was overlapped by new intensive bands at 1112 cm^−1^, 1110 cm^−1^, and 1044 cm^−1^ (assigned to stretching C–O and O–H vibrations) from BSA–cyclitol complexes [69]. The additional peaks at 998, 996, and 994 cm^−1^ in the protein complexes can be assigned to stretching C–O and C–C vibrations [53,70]. These results confirm the sorption process of cyclitols accompanied by an increase in the availability of hydroxyl groups on the protein surface.

The shift in the amide A, amide I, and amide II bands indicated the cyclitol interactions with the polypeptide backbone. Furthermore, ligands are involved in the formation of new hydrogen bonds with the macromolecule; nevertheless, the strength of these interactions is weak compared with a protein–water complex, which leads to a less polar environment for BSA. New peaks in the range of the amide III band displayed a slight change in the secondary structure content after cyclitol addition. In addition, it was noticed that D-sorbitol, adonitol, and D-(–)-quinic acid interacted with the histidine side chain. The shift in the peak assigned to the Asp and Glu side chains to higher frequencies confirmed the enhancement of the hydrophobicity of the protein environment in the presence of cyclitols.

### 2.6. Raman Spectroscopy

Raman spectroscopy is an essential technique in the analysis of a protein and allows for the monitoring of conformational changes in its secondary and tertiary structures. Raman spectra are more suitable for amide band observation than FTIR analysis due to fluorescence phenomena [71]. In Figure 7, the Raman spectra of BSA, the cyclitols, and their complexes are shown in the fingerprint region of 1700–900 cm^−1^. A detailed list of the captioned wavenumbers in Figure 7 is presented in the Appendix A. The most significant changes were observed in the BSA/D-sorbitol spectra. New bands were observed at 1604, 1349, 1234, and 1178 cm^−1^ after the addition of D-sorbitol.

The amide I band (1700–1600 cm^−1^) mainly includes stretching C=O vibrations [55]. Based on the weak intensity of this band for the analyzed samples, it is suggested that Raman spectroscopy is a less-sensitive technique in this range, which may be related to fluorescence interference [62]. The new band at 1604 cm^−1^ for the BSA–D-sorbitol complex could be associated with the formation of hydrogen bonds between the tyrosine side chain and the cyclitol [72,73].

The bands at 1568–1567 cm^−1^ in the BSA–cyclitol complexes originate from asymmetric stretching vibrations of deprotonated carboxyl groups COO^−^ of glutamic and aspartic acids, indicating the interaction of cyclitols and amino acids with acidic side chains [39]. These signals could be attributed to deprotonated carboxylic groups and allow us to confirm the mechanism of cyclitol sorption leading to a decrease in the pH of the solution.

The signals at 1511, 1509, 1508, and 1504 cm^−1^ are characteristic of tryptophan indole ring interactions with cyclitols (C–N stretching vibrations coupled with C–H and N–H bending vibrations) and confirm that cyclitols are in close proximity to the tryptophan residue [59].

The signal at 1456 cm^−1^ (responsible for CH_2_ scissor vibrations) appears in the presence of D-sorbitol [53]. The peaks at 1415, 1414, and 1412 cm^−1^ for protein complexes (assigned to stretching C–N vibrations of glycine) indicate cyclitol interactions with glycine residues [52]. BSA–D-sorbitol interactions induced the appearance the new bands at 1349 cm^−1^, resulting in bending C–H vibrations of tryptophan residues [72,74].

The amide III band includes the range 1315–1240 cm^−1^ and is characteristic of stretching C–N and in-plane N–H bending vibrations. The obtained signals in this range allow us to determine changes in the protein secondary structure. The recorded signals at 1309, 1308, and 1307 cm^−1^ are responsible for the α-helix type of secondary structure [75]. The addition of D-sorbitol caused the appearance a new band at 1234 cm^−1^ (stretching C–O coupled with C–C vibrations of tyrosine). This proves that hydrogen bonds were formed, where tyrosine acts as a hydrogen bond acceptor. Another new band for the BSA–D-sorbitol sample was observed at 1178 cm^−1^ that includes exocyclic C–H and in-plane O–H bending vibrations of the tyrosine residue. This band may be related to a shift in the O–H group outside the ring plane, which is directly connected to the formation of hydrogen bonds [76].

The intensive band for BSA at 1098 cm^−1^ can be assigned to stretching C–N coupled with bending C–H vibrations of histidine. This band was slightly shifted to lower frequencies for the BSA–cyclitol complexes, resulting in overlapping with the band at 1046 cm^−1^ that originates from cyclitol stretching C–O vibrations [74]. The new peak for the BSA–D-sorbitol sample at 987 cm^−1^ assigned to stretching C–O vibrations of serine indicates cyclitol interactions with the serine side chain [52].

Based on the obtained results, it is suggested that cyclitols are engaged in the sorption process onto the protein. All of the analyzed cyclitols participated in binding with deprotonated glutamic and aspartic acid side chains of BSA, which allows us to confirm the mechanism of interactions obtained from the kinetic study. All of the analyzed cyclitols were involved in interactions with histidine side chain and glycine residues. New bands appeared in the presence of D-sorbitol, indicating its participation in the formation of new intermolecular hydrogen bonds with tryptophan, tyrosine, and serine side chains. Such a tendency of D-sorbitol for the creation of hydrogen bonds could be related to the higher number of available hydroxyl groups in its structure [77]. The appearance of new peaks in the presence of D-sorbitol could be induced by partial self-association of cyclitol molecules via hydrogen bonding [78]. D-sorbitol clustering leads to a weakening of cyclitol–water interactions; however, it increases the intermolecular interactions between solvent molecules. Thus, the preferential interactions of D-sorbitol with BSA particles stand out compared with other cyclitols. Based on the obtained results, Raman spectroscopy is less sensitive for the analysis of changes in the secondary structure compared with FTIR spectroscopy; nevertheless, it provides a detailed analysis of cyclitol interactions with protein side chains. Thus, these two techniques are complementary to each other.

### 2.7. Isothermal Titration Calorimetry

Figure 8 presents the titration curves obtained for sorbitol and quininc acid. Binding at low specificity is hard to monitor by means of calorimetry due to the low enthalpy contribution, which is the basis for detection in this method. We could not observe any changes in enthalpy for shikimic acid and for adonitol. For sorbitol, the obtained curve is almost linear, so an unambiguous determination on the binding parameters is difficult to achieve (the error is almost equal to the value). We conclude that the binding of quinic acid is more specific and stronger than that of sorbitol with the following binding parameters: N = ~5, K_D_ = 159 ± 113 µM, ΔH = –418 ± 125, –TΔS = −4.8. We also observed that in the case of sorbitol, the binding is endothermic whereas in the case of quinic acid it is exothermic with a negative enthalpy change. The calculated values of changes in the Gibbs free energy are complementary to those obtained from the kinetic studies and molecular modeling. The binding of cyclitols to BSA should release the water molecules from the protein surface, increasing the entropy of the system. In the case of quinic acid, it possesses a lower number of hydroxylic groups but one carboxylic group in comparison with D-sorbitol. Thus, the binding of quinic acid should cause the desorption of a lower number of water molecules from the protein surface but the occurrence of electrostatic interactions between the cyclitol and BSA. Thus, it is reflected by the respective thermodynamic parameters. Moreover, as can be seen from the kinetic studies, the D-sorbitol and adonitol experienced a delay in the adsorption in comparison with the D-(–)-quinic acid and shikimic acid, which, among other things, may be connected to the heat effects of the reaction.

### 2.8. Molecular Docking

Molecular docking is an extremely useful bioinformatics tool that allows us to predict the site and nature of interactions between proteins and attached low-molecular-weight ligands. In the present study, BSA was used as a receptor molecule, while ligands were represented by cyclitol molecules. The main idea of a simulation is based on finding the strongest bond between at least two unbound structures. Currently, simulations can be performed by a number of specially prepared programs; however, AutoDock Vina is by far the best in terms of the accuracy of the results obtained in relation to the time spent [79]. This software is extremely easy to use and allows laymen to model appropriate structures by, among other things, automatically adjusting the grid map and grouping the results [80].

Figure 9 displays the potential binding sites between the BSA protein receptor and cyclitol ligands based on the molecular docking analysis. The calculated binding energies for all complexes are shown in Table 4.

The calculated binding energy of D-sorbitol to BSA was −5.5 kcal/mol. Hydrogen bonds were formed between D-sorbitol and the following amino acids: Arg458, Asp103, and His145. The lowest binding energy of protein–cyclitol interactions (−5.0 kcal/mol) was noticed in the case of adonitol. The following amino acid residues were involved in the formation of the BSA–adonitol complex: Arg458, Ser192, and His145. The binding energy of shikimic acid to BSA was −5.7 kcal/mol. The side chains of two arginine residues (Arg217 and Arg256) interacted with shikimic acid by hydrogen bonding. The strongest interaction was observed for BSA and D-(–)-quinic acid. The calculated binding energy was −6.0 kcal/mol. Cyclitol complexed with the protein by the following amino acids: Arg217, Arg198, Ser191, and His286. Interestingly, hydrogen bonding interactions between aspartic acid, histidine, and serine residues and cyclitols also were confirmed with the observed bands in the FTIR and Raman spectra.

## 3. Materials and Methods

### 3.1. Materials

Bovine serum albumin (≥99% purity), D-sorbitol (99% purity), adonitol (≥99% purity), shikimic acid (≥99% purity), D-(***–***)-quinic acid (98% purity), and formic acid were purchased from Sigma-Aldrich (St. Louis, MO, USA). Isopropanol, acetonitrile and water (LC-MS grade) were obtained from Supelco (Bellefonte, PA, USA).

### 3.2. Preparation of BSA Solution

Bovine serum albumin powder was dissolved in an Eppendorf tube to a concentration of 1 mg/mL.

### 3.3. Preparation of Cyclitol Solutions

The molar concentration of the prepared stock solutions for D-sorbitol, adonitol, shikimic acid, and D-(***–***)-quinic acid was 3.01 × 10^−2^ M, which is equal to 5.48 mg/mL, 4.58 mg/mL, 5.24 mg/mL, and 5.78 mg/mL, respectively.

### 3.4. Sample Preparation for SDS-PAGE Separation

Protein control and protein–cyclitol samples (at a molar ratio of 1:100) were prepared in accordance with the Bis-Tris Plus Mini Gels Protocol. Electrophoresis was performed at a constant voltage (200 V) for 22 min. The opened gel was stained in Coomassie Brilliant Blue (1 h); then, it was placed into water for de-staining (24 h).

### 3.5. Kinetic Study of Cyclitol Binding to BSA

At the beginning of the experiment, the pH of the BSA solution (0.67 mg/mL) was measured with the help of a pH meter (METTLER TOLEDO FiveEasy Plus, Greifensee, Switzerland) at 22 °C. After that, to four separate falcones with 2 mL of BSA solution (1 mg/mL) was added 1 mL of cyclitol solution to obtain the protein–cyclitol complex at a 1:10 molar ratio. All solutions were mixed with a test tube shaker (Vortex-Genie 2, Bohemia, NY, USA). pH measurements of the final solutions were conducted 1, 5, 10, 30, 60, 120, 300, and 1440 min after the cyclitol’s addition. Additionally, the pH of the cyclitol control solutions was also determined. Molecular recognition is very complicated process and can be described in terms of selective non-specific interactions of a macromolecule (for example, a target protein) with a ligand (3), providing the formation of a complex [81]:(3)$$P+L\;\overset{\leftarrow }{\to }\mathrm{PL}$$
where P is the protein, L is the ligand, and PL is the protein–ligand complex.

In the present work, a zero-order model was used to determine the rate constant of each stage of an interaction. Additionally, the Weber–Morris intraparticle diffusion model was used to explain the mechanism of interactions between BSA and cyclitols.

The rate of a zero-order reaction (4) is represented by the following equation [82]:(4)$$v=\;-\frac{\mathrm{dc}}{\mathrm{dt}}=k_{0},$$
where *v* is the rate of the zero-order reaction ((mg/L)/min); $\frac{\mathrm{dc}}{\mathrm{dt}}$ is the change in the concentration over time ((mg/L)/min); and k_0_ is a first-order rate constant ((mg/L)/min). In our case, the change in pH over time was measured instead of the change in concentration (5); thus, the equation above was transformed to:(5)$$v=\;-\frac{\mathrm{dpH}}{\mathrm{dt}}=k_{0},$$

The zero-order reaction kinetic model (6) can be described by the following formula:pH = pH_0_ − k_0_t,(6)
where pH is the acidity of the protein–cyclitol solution after a certain period of time; pH_0_ is the initial pH of the BSA solution (0.67 mg/mL); and t is the amount of time that passed after the addition of the cyclitol to the protein solution (min).

Protein–cyclitol interactions can be accompanied by a change in the pH of the solution. The Weber–Morris intraparticle diffusion model (7) allows for the representation of the mechanism of cyclitol sorption based on the change in the availability of hydroxyl groups in the BSA protein [39]:q_t_ = A + K_ip_t^0.5^,(7)
where q_t_ is the change in the number of hydroxyl groups on the BSA at time t (mg/g), A is a constant that expresses the amount of external surface adsorption or boundary layer diffusion (mg/g), and K_ip_ is the diffusion rate constant ((mg/g)/t^0.5^).

Formation of a new biocolloidal system (BSA–cyclitol complex) provides a different distribution of hydroxyl ions in the dispersed phase and the dispersion medium. The distribution constant K_D_ within the equilibrium time (8) can be calculated using the following ratio [42]:(8)$$K_{D}=\frac{q}{\mathrm{Ce}},$$
where q is the change in the number of hydroxyl groups on the protein surface resulting in interactions with cyclitols (mg/kg); and Ce is the equilibrium concentration of hydroxyl ions in the solution (mg/L).

One of the most import thermodynamic parameters is the Gibbs free energy (thermodynamic potential), which allows us to characterize the spontaneity of the binding process (9) as well as the stability of the protein–cyclitol complex [81]:∆G_bind_ = −RT ln(K_D_),(9)
where ∆G_bind_ is the binding free energy in Joules; R is the universal gas constant (8.314 J/K×mol); and T is the Kelvin temperature. More negative values of the binding free energy and higher values of the binding affinity constant prove the formation of a more stable protein–ligand complex [81]. The formula above allows for the determination of the spontaneity of interactions between BSA and different cyclitols.

### 3.6. HPLC-MS/MS Analysis

Qualitative profiling and quantitative profiling of cyclitols and their complexes with BSA were performed using high-pressure liquid chromatography coupled with mass spectrometry.

#### 3.6.1. Analysis of Protein–Cyclitol Interactions

An HPLC-MS system (LCMS-8050 SHIMADZU, Kyoto, Japan) equipped with an ESI source and a triple quadrupole analyzer was used. Cyclitols were analyzed using a column packed with BSA particles (RESOLVOSIL BSA-7, Macherey-Nagel, Düren, Germany) at 30 °C. Analysis of BSA–cyclitol interactions was performed with the use of eight mobile phases composed of water, isopropanol, and formic acid. The mass spectrometry parameters were as follows: ion source temperature, 230 °C; flow rate, 0.5 mL/min; nebulizing gas, 2 L/min; capillary voltage, 4 kV. Scans were recorded in the range (*m/z*) 50 to 400.

#### 3.6.2. Indication of Unbounded Cyclitols

Calibration of each cyclitol was performed in the following concentration range: 1 × 10^−6^–1 × 10^−5^ M. Samples for the determination of unbounded cyclitols were prepared in the following way: Mixtures of BSA (200 µL) and cyclitol (100 µL) at the appropriate concentrations were incubated for 1 h in a thermomixer (Eppendorf Comfort, Hamburg, Germany) at 600 rpm and 20 °C. After that, the tubes were centrifuged at 14,000 rpm for 10 min. The obtained supernatant was diluted 30 times. Quantitative analysis was carried out without a column. The amount of free cyclitol and the unbounded fraction were determined in SIM mode. The mobile phase was composed of 0.1% *(v/v)* formic acid–water (A) and acetonitrile (B) (75:25 *(v/v)*).

### 3.7. Spectroscopic Analysis

#### 3.7.1. UV-VIS Spectroscopy

UV-VIS spectra for 1.0 mg/mL of BSA solution and a 3.01 × 10^−2^ M solution of each cyclitol were obtained by a NanoDrop 2000c UV-VIS Spectrophotometer (Thermo Scientific, Waltham, MA, USA) in the 190–840 nm range of excitation. Samples were analyzed in a quartz cuvette with a path length of 1 cm at 20 °C. 

#### 3.7.2. Fluorescence Spectroscopy

Three-dimensional fluorescence spectra of native BSA and BSA complexes were recorded with an FP-8300 Spectrofluorometer (Jasco, Pfungstadt, Germany). BSA–cyclitol complexes at a 1:1000 molar ratio were prepared by mixing a 1 mg/mL protein solution with a 3.01 × 10^−2^ M cyclitol solution. The samples were incubated for 1 h in a swivel roller mixer (Paul Marienfield, Lauda-Königshofen, Germany). In the next step, the solutions were transferred to an Amicon Ultra-4 centrifugal filter and centrifugated for 20 min at 8000 rmp and 4 °C. Unbounded cyclitol was removed after washing the solutions with water two times. The remaining protein–cyclitol solutions were frozen and lyophilized by a freeze dryer (LABCONCO FreeZone, Kansas City, MO, USA) for 24 h. Dried complexes were dissolved in water in order to obtain solutions with a final concentration of BSA of 0.025 mg/mL. The samples were analyzed in a quartz cuvette with a 1-cm path length. Spectra measurements were performed in the 210–750 nm range of excitation and the 200–735 nm range of emission wavelengths, respectively, with a 5 nm interval and a 5 nm/min scan speed.

#### 3.7.3. Fourier Transform Infrared Spectroscopy (FTIR)

A total of 200 µL of BSA solution and 100 µL of cyclitol solution were added to an Amicon Ultra-0.5 centrifugal filter in order to obtain a molar ratio of 1:10. A protein control solution was prepared by mixing 200 µL of BSA solution with 100 µL of water. The BSA–cyclitol samples were incubated for 1 h in a thermomixer (Eppendorf Comfort, Hamburg, Germany) at 600 rpm and 20 °C to allow for the complex’s formation. After that, the tubes were centrifuged at 14,000 rpm for 10 min. The remaining solution containing the protein–cyclitol complex in the filtration part was washed with 300 µL of water and centrifuged again under the same conditions. The infrared spectra of the BSA control and BSA–cyclitol complex samples were measured with an FTIR spectrometer (Perkin Elmer, Waltham, MA, USA) in the MIR range (4000–900 cm^−1^) with the use of a CaF_2_ optical window and the thin layer method. Data were collected with 20 scans and the resolution was set to 8 cm^−1^. Spectra were processed using the OriginPro 2016 software.

#### 3.7.4. Raman Spectroscopy

BSA–cyclitol complex samples were prepared in the same way as for the FTIR analysis. After washing, protein–cyclitol solutions from the filtration part were removed, put in Eppendorf tubes, protected with Parafilm, and placed in a freezer for 5 min. Then, probes were lyophilized with a freeze dryer (LABCONCO FreeZone, Kansas City, MO, USA) for 3 h. Dried samples were spotted onto a microscope slide and mixed with 2 µL of water. Crystalline BSA powder was used in the case of the protein control sample. The prepared samples were analyzed with a spectrometer (Bruker Optics, Ettlingen, Germany) under the following conditions: laser wavelength, 532 nm; power, 20 mW; magnification, 20 times; and exposure time, 60 s. Spectra were processed using the OriginPro 2016 software.

### 3.8. Isothermal Titration Calorimetry

Lyophilized protein was dissolved in phosphate buffer (30 mM, pH 8.0) and dialyzed against this buffer overnight. Ligands were dissolved in the dialysis buffer to a final concentration of 5 mM. Titrations were performed at 25 °C using a calorimeter (Malvern MicroCal PEAQ-ITC, Malvern, UK). The protein in the cell was kept at an ~110 µM concentration (determined at 280 nm) and titrated with aliquots of 2 µL of 5 mM ligand (D-(–)-quinic acid, shikimic acid, D-sorbitol, or adonitol). The raw ITC data were analyzed with the MicroCal-ITC Analysis Software to obtain the thermodynamic parameters of the complexation reaction (stoichiometry (N), dissociation constant (K_D_), and changes in enthalpy (ΔH) and entropy (ΔS)).

### 3.9. Molecular Docking

The free and trial Autodock software was used to simulate the attachment of the cyclitols to the BSA. The 3D structure of the protein in the .pdb extension was downloaded from https://www.rcsb.org (accessed on 17 February 2022). Three-dimensional structures of ligands in the .sdf extension were downloaded from https://pubchem.ncbi.nlm.nih.gov (accessed on 17 February 2022). Open Babel, which was used to convert files, was downloaded from http://openbabel.org (accessed on 17 February 2022). The analysis was conducted and visualized by the AutoDockTools software, which can be downloaded at https://autodock.scripps.edu. AutoDock Vina (available at https://github.com/ccsb-scripps/AutoDock-Vina/releases/tag/v1.2.3 (accessed on 17 February 2022)) was used for the process of attaching the ligand to the protein. Additionally, Chem3D Pro 12 trial version, https://cs-chem3d-std.software.informer.com/4.0 (accessed on 17 February 2022), and Molegro Molecular Viewer 7, http://molexus.io/molegro (accessed on 17 February 2022), were used to optimize the 3D structures of the protein and ligands. The PyLol program was used to visualize the obtained results (https://pymol.org/2/ (accessed on 17 February 2022)).

The 3D structure of the BSA (4f5s) protein (receptor) was downloaded from the RCSB Protein Data Bank website. The file was in .pbd format. The downloaded file contained a two-chain protein structure, an attached ligand (TRIETHYLENE GLYCOL), and a water molecule as a solvent. Using the Molegro Molecular Viewer 7 software, the water molecule and ligand were removed from the file. Due to the identity of both chains, only the A chain was used for further simulations in order to reduce the time needed to perform the calculations. Using the AutoDockTools program, we checked whether the isolated chain had lost atoms. Polar hydrogens and merged non-polar hydrogens were then added. Finally, Kollman charges were added. The prepared protein was saved in .pdbqt format.

The 3D structures of D-sorbitol, adonitol, shikimic acid, and D-(–)-quinic acid were downloaded from the PubChem website. Then, the minimization of energy values for all ligands was performed by the Chem3D Pro program. At the end of the process, the files were saved in .pdb format and opened in AutoDockTools for further preparation. After loading the ligand file, the program automatically added Gesteiger charges, merged non-polar hydrogens, and detected rotational bonds. We only needed to detect and mark the torsion tree. The prepared ligand was saved in .pdbqt format, allowing it to be linked to the previously prepared protein. The conf .txt file was prepared in such a way that all the center axes and size axes of the set grid were written with the receptor, ligand, and output file extensions.

The docking simulation was performed using the AutoDock Vina executable file. The procedure for molecular docking between the receptor and the ligand was described by Trott et al. (2009) (O. Trott and A. J. Olson, “AutoDock Vina: Improving the Speed and Accuracy of Docking with A New Scoring Function, Efficient Optimization, and Multithreading,” *J. Comput. Chem.*, vol. 31, no. 2, 2009) and Vina et al. (2020) (D. Vina, H. S. Albumin, S. Octanoate, and A. Vina, “How to Perform Docking in A Specific Binding Site Using AutoDock Vina?,” pp. 1–13, 2020). The results were visualized in AutoDockTools.

## 4. Conclusions

In the present work, we investigated the molecular mechanism of interactions between BSA as a model protein and four types of cyclitols (D-sorbitol, adonitol, shikimic acid, and D-(–)-quinic acid). Initially, using SDS-PAGE electrophoresis, it was determined that none of the analyzed cyclitols led to the destabilization of the BSA structure compared with the control sample. The zero-order kinetic model demonstrated that the sorption process of D-sorbitol, adonitol, and D-(–)-quinic acid had a non-homologous character, whereas the formation of the BSA–shikimic acid complex was accompanied by a rapid one-step sorption process. The Weber–Morris intraparticle diffusion model indicated that the decrease in the amount of cyclitol in the solution was mainly associated with binding on the external protein surface. Nevertheless, the linear cyclitols (D-sorbitol and adonitol) partly underwent diffusion into the protein structure, probably because of their more flexible structure. The binding process of all cyclitols was regulated by non-specific weak interactions and occurred in a spontaneous way. Cyclic-structure molecules (shikimic acid and D-(–)-quinic acid) were involved in much stronger interactions with the protein particle than linear cyclitols (D-sorbitol and adonitol), which was confirmed by the higher values of the retention time from the HPLS-MS/MS analysis and the more effective quenching of protein fluorescence. Fluorescence spectroscopy also showed that cyclitols do not contribute to significant changes in the polarity in the aromatic amino acid microenvironment. FTIR and Raman spectroscopy confirmed that the sorption mechanism of cyclitols was mediated by aspartic and glutamic acids. Moreover, the shift in the amide bands in FTIR spectra indicated the cyclitol interactions with the polypeptide backbone. Nevertheless, these interactions were considered to be weaker compared with the protein–water interactions, leading to a less-polar environment for BSA. New peaks observed in the FTIR and Raman spectra were assigned to the formation of new intermolecular hydrogen bonds between cyclitols and amino acid side chains, such as histidine, glycine, serine, cysteine, tryptophan, tyrosine, and phenylalanine. For the establishment of a more detailed mechanism of interaction between the protein and cyclitols, molecular modeling was performed. The determined potential binding sites were partially adjusted with observed bands in the FTIR and Raman spectra. Additionally, the obtained values of binding energies were close to the experimental results, which allows us to assume that the proposed binding site is consistent with reality. Moreover, ITC measurements were performed, indicating changes in entropy but not in heat effects.

## Figures and Tables

**Figure 1 ijms-23-02940-f001:**
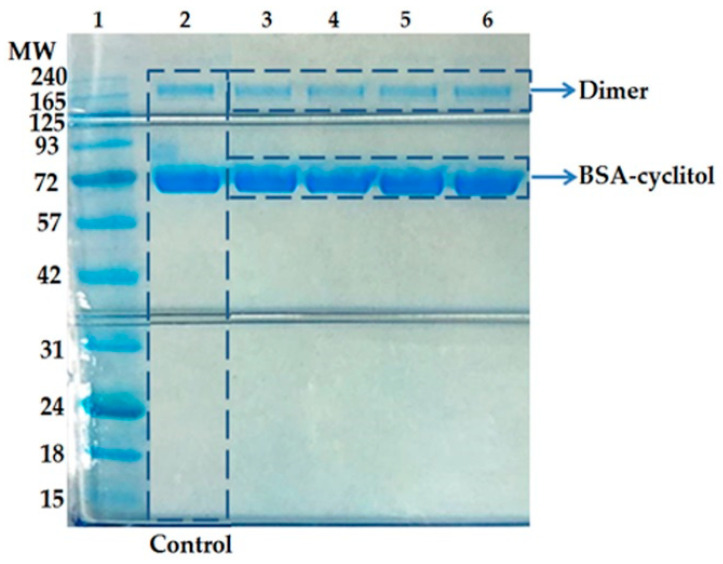
SDS-PAGE electropherogram: 1, BlueStar Plus Prestained Protein Marker; 2, BSA (control); 3, BSA/D-sorbitol; 4, BSA/adonitol; 5, BSA/shikimic acid; 6, BSA/D-(***–***)-quinic acid.

**Figure 2 ijms-23-02940-f002:**
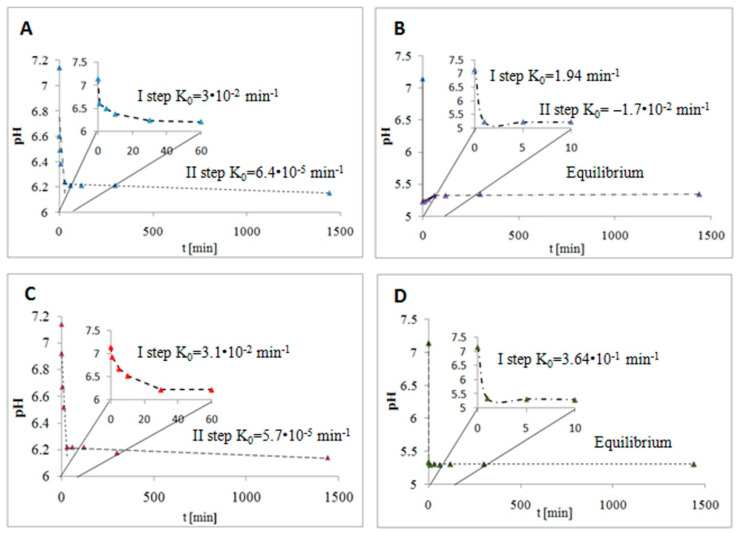
Zero-order kinetic model of protein–cyclitol interactions (at the 1:10 molar ratio): (**A**) BSA–adonitol; (**B**) BSA–D-(***–***)-quinic acid; (**C**) BSA–D-sorbitol; (**D**) BSA–shikimic acid.

**Figure 3 ijms-23-02940-f003:**
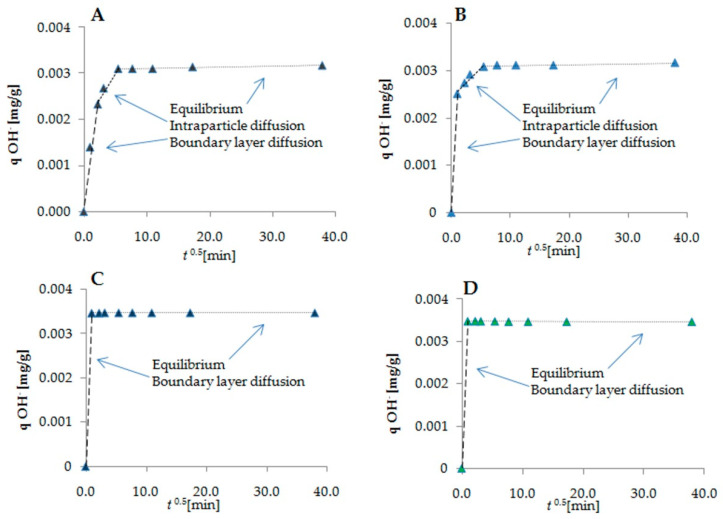
Weber–Morris intraparticle diffusion model of cyclitol sorption: (**A**) D-sorbitol; (**B**) adonitol; (**C**) shikimic acid; (**D**) D-(***–***)-quinic acid.

**Figure 4 ijms-23-02940-f004:**
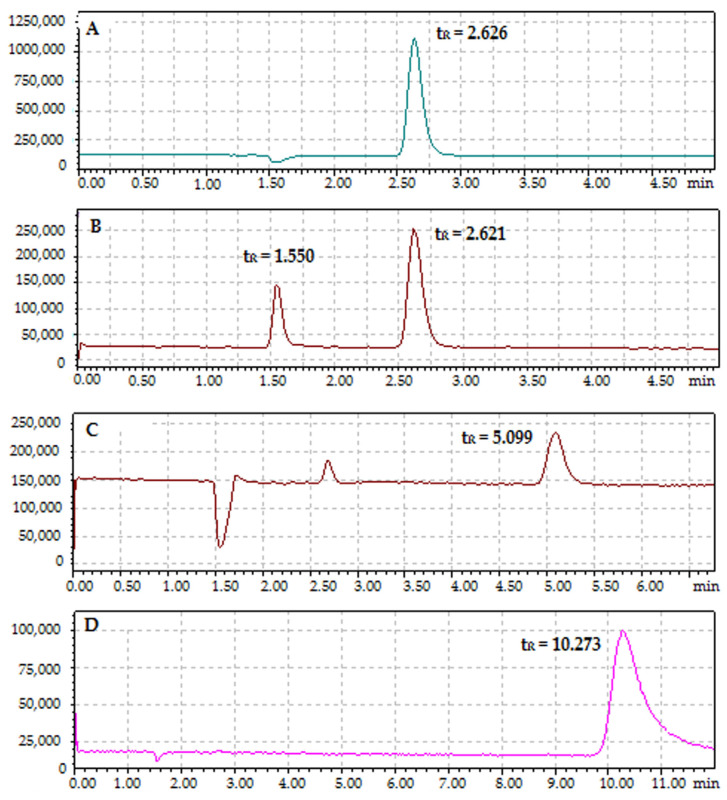
Chromatograms of: (**A**) adonitol; (**B**) D-sorbitol; (**C**) shikimic acid; and (**D**) D-(***–***)-quinic acid (under the following conditions: stationary phase, column filled with BSA; mobile phase, water + 0.1% *v/v* FA).

**Figure 5 ijms-23-02940-f005:**
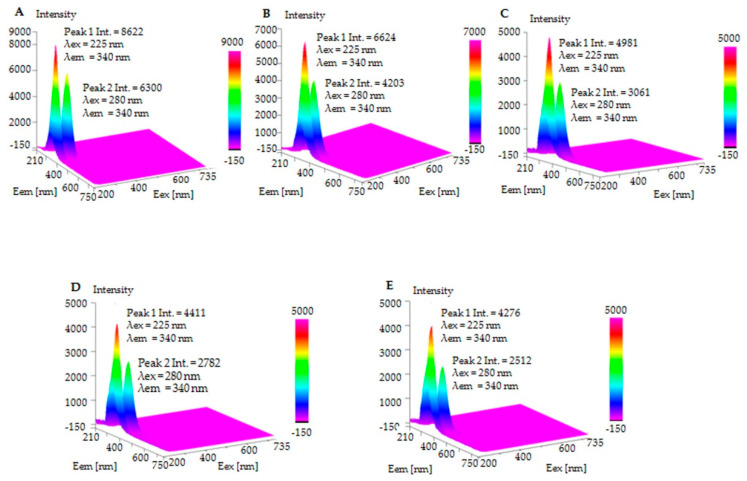
The 3D-fluorescence spectra of: (**A**) BSA; (**B**) D-sorbitol–BSA complex; (**C**) Adonitol–BSA complex; (**D**) Shikimic acid–BSA complex; and (**E**) D-(***–***)-quinic acid–BSA complex.

**Figure 6 ijms-23-02940-f006:**
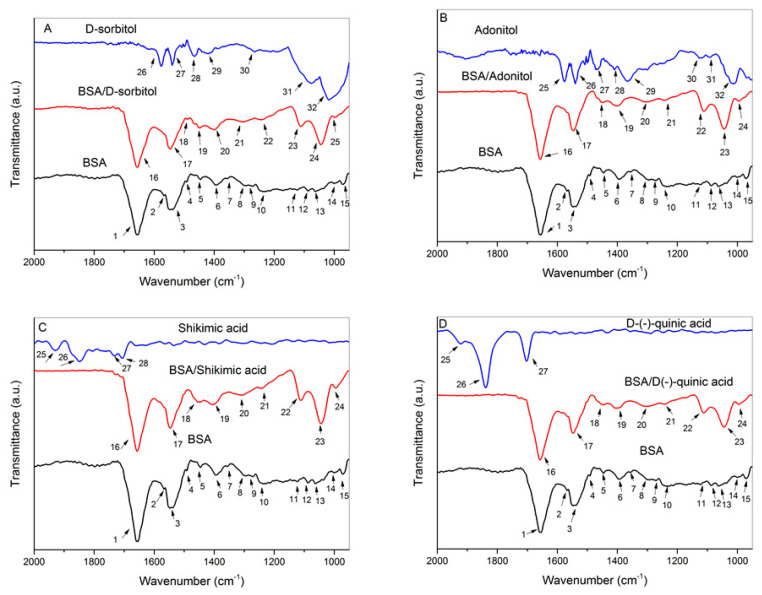
FTIR spectra of BSA, the cyclitols, and their complexes (at a 1:10 molar ratio) in the range of 2000–950 cm^−1^: (**A**) D-sorbitol; (**B**) adonitol; (**C**) shikimic acid; (**D**) D-(–)-quinic acid.

**Figure 7 ijms-23-02940-f007:**
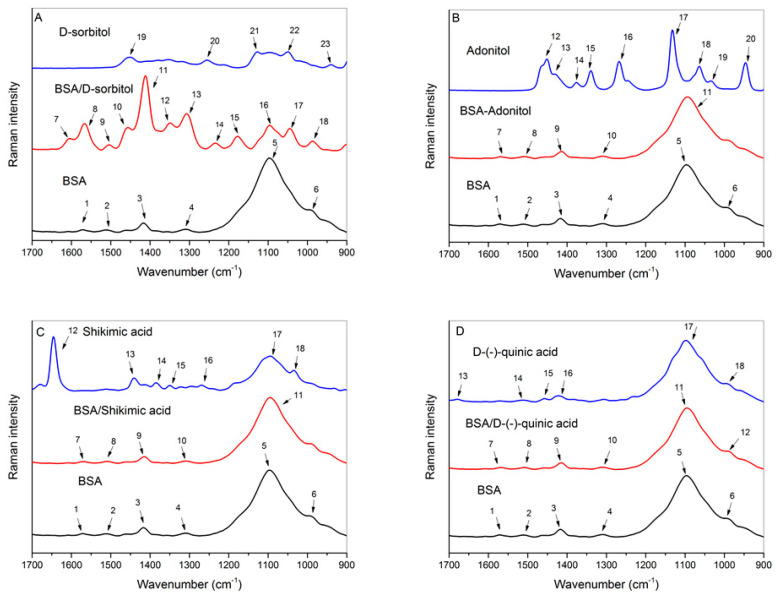
Raman spectra of BSA, the cyclitols, and their complexes (at a 1:10 molar ratio) in the range of 1700–900 cm^−1^: (**A**) D-sorbitol; (**B**) adonitol; (**C**) shikimic acid; and (**D**) D-(–)-quinic acid.

**Figure 8 ijms-23-02940-f008:**
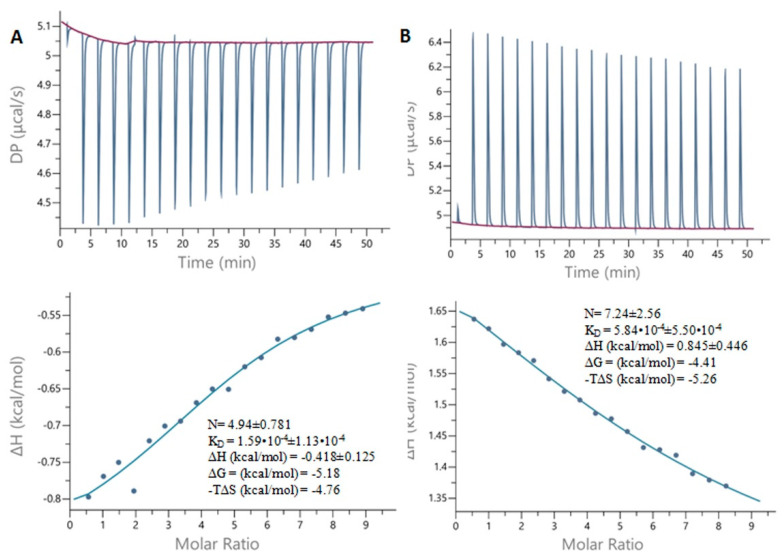
The results of ITC measurements of (**A**) D-(–)-quinic acid–BSA; (**B**) D-sorbitol–BSA.

**Figure 9 ijms-23-02940-f009:**
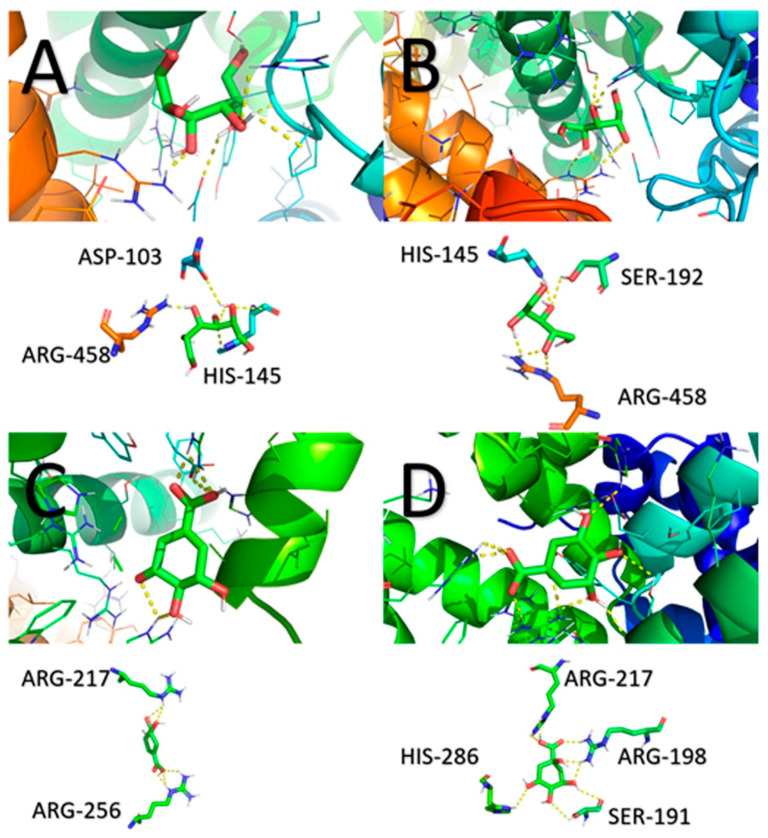
Potential binding sites of BSA with: (**A**) D-sorbitol; (**C**) adonitol; (**B**) shikimic acid; and (**D**) D-(***–***)-quinic acid.

**Table 1 ijms-23-02940-t001:** Parameters of kinetic models and Gibbs free energy change values of BSA–cyclitol interactions.

Cyclitol	Zero-Order Kinetic Model		Intraparticle Diffusion Model		Gibbs Free Energy Change ΔG (kJ/mol)/(kcal/mol)
D-sorbitol	Step I	0.031	A	0.00199	−21.74/−5.20
	k_0_ (min^−1^)		(mg/g)		
	Step II	0.000057	K_ip_	0.0000477	
	k_0_ (min^−1^)		((mg/g)/t^0.5^)		
Adonitol	Step I	0.030	A	0.00230	−21.69/−5.18
	k_0_ (min^−1^)		(mg/g)		
	Step II	0.000064	K_ip_	0.0000352	
	k_0_ (min^−1^)		((mg/g)/t^0.5^)		
Shikimic acid	Step I	0.364	A	0.00281	−26.37/−6.30
	k_0_ (min^−1^)		(mg/g)		
			K_ip_	0.0000289	
			((mg/g)/t^0.5^)		
D-(*–*)-quinic acid	Step I	1.94	A	0.00282	−26.16/−6.25
	k_0_ (min^−1^)		(mg/g)		
	Step II	−0.017	K_ip_	0.0000285	
	k_0_ (min^−1^)		((mg/g)/t^0.5^)		

**Table 2 ijms-23-02940-t002:** Retention times of analyzed cyclitols at different mobile phases.

Composition of Mobile Phase	t_R_								
	Adonitol *m/z* (+) = 153			D-Sorbitol *m/z* (–) = 181		Shikimic Acid *m/z* (–) = 173		D-(*–*)-Quinic Acid *m/z* (–) = 191	
	[M − H]^+^	[M − Na]^+^	[2M − Na]^+^	[M]^−^	[2M]^−^	[M]^−^	[2M]^−^	[M]^−^	[2M]^−^
Water	2.645	2.641	-	1.516	-	-	-	-	-
				2.638					
Water + 0.1% *v/v* FA	2.626	2.626	-	1.550	1.550	5.099	-	10.273	-
				2.621					
Water + 0.5% *v/v* IP	2.792	2.788	2.786	1.726	2.821	-	-	-	1.769
				2.822					
Water + 1% *v/v* IP	2.903	2.900	-	1.672	2.888	-	-	-	1.721
				2.887					
Water + 2% *v/v* IP	2.989	2.989	-	1.656	2.983	-	-	-	1.721
				2.983					
Water + 0.1% *v/v* FA + 0.5 % *v/v* IP	2.630	2.630	-	1.490	-	5.064	-	10.269	-
				2.623					
Water + 0.1% *v/v* FA + 1% *v/v* IP	2.641	2.638	2.632	1.559	1.561	5.060	-	10.297	-
				2.631					
Water + 0.1% *v/v* FA + 2% *v/v* IP	2.635	2.635	2.635	1.551	1.554	5.057	-	10.344	-
				2.644					

FA, formic acid; IP, isopropanol; [M − H]^+^ and [M]^−^ are pseudomolecular ions of adonitol and the rest of the cyclitols, respectively; [2M]^−^ is the dimer ion of the cyclitols; [M − Na]^+^ is the sodium adduct of the monomer and [2M − Na]^+^ is the sodium adduct of the dimer (for adonitol).

**Table 3 ijms-23-02940-t003:** Number of bounded cyclitol molecules based on the initial protein–cyclitol molar ratio calculated by the equation obtained from the calibration curve.

Analyzed Cyclitol	*m/z*	Protein–Cyclitol Molar Ratio	Bounded Cyclitol with Protein ± SD	Equation of the Calibration Curve y = ax + b (R^2^)	Number of Bounded Ligand Molecules ± SD
D-sorbitol	[M]^−^ = 181	1:2	49.92 ± 0.29	y = 4.00 × 10^11^ + 506,935 (0.9985)	0.99 ± 0.01
		1:5	17.75 ± 1.02		0.89 ± 0.01
		1:7	23.92 ± 1.13		1.67 ± 0.08
		1:10	20.89 ± 1.25		2.09 ± 0.13
Shikimic acid	[M]^−^ = 173	1:2	92.56 ± 1.49	y = 1.29 × 10^11^ + 85,318 (0.9982)	1.85 ± 0.03
		1:5	2.75 ± 0.09		54.98 ± 1.77
		1:7	3.20 ± 0.02		45.70 ± 0.31
D-(*–*)-quinic acid	[M]^−^ = 191	1:2	3.95 ± 0.44	y = 5.00 × 10^11^ + 433,847 (0.9960)	0.08 ± 0.01
		1:7	32.55 ± 1.31		2.28 ± 0.09
		1:10	39.27 ± 0.94		3.93 ± 0.09

**Table 4 ijms-23-02940-t004:** Predicted cyclitol binding sites and calculated binding energies.

Protein Receptor	Ligand	Amino Acid Residues	Number of Bonds	Bond Length (A°)	Calculated Binding Energy of the First Ligand Molecule to BSA (kcal/mol)
BSA	D-sorbitol	Arg458	2	3.2/1.9	
		His145	1	2.8	−5.5
		Asp103	1	2.4	
BSA	Adonitol	Arg458	3	1.2/2.1/2.5	
		Ser192	1	2.4	−5.0
		His145	1	2.0	
BSA	Shikimic acid	Arg217	2	2.6/2.8	−5.7
		Arg256	3	2.1/2.5/2.3	
BSA	D-(–)-quinic acid	Arg217	2	2.3/2.4	
		Arg198	3	2.1/2.1/2.3	−6.0
		Ser191	2	2.9/2.6	
		His286	2	3.3/2.8	

D-(***–***)-quinic acid and shikimic acid had a higher binding affinity for BSA than linear molecules as also shown by the HPLC-MS/MS analysis. The molecular docking results are close to the experimental values of binding energy obtained from the kinetic study. It can be assumed that the proposed binding site is consistent with reality.

## Data Availability

No applicable.

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
