# Peer review of "The Study of Protein–Cyclitol Interactions"

_ijms, 2022, doi:10.3390/ijms23062940_

Round 1

Reviewer 1 Report

This paper discusses the interactions of four natural products with bovine serum albumin. They have used a number of methods/techniques to investigate the interaction. The topic of study is interesting and this could be a good paper but in my opinion there are so many flaws both in writing and analyses. The English of the manuscript also need through revision. Fluorescence studies have not been carried out accurately and need to be repeated or omitted from the paper. My comments to the authors are as follows:

  1. The importance of working with cyclitols should be elaborated in the introduction section. Why do the authors think that the study of these compounds with Bovine serum albumin would be of interest? Further, the description and importance of each cyclitol must be given in the introduction section. Overall introduction section should be rewritten.
  2. Authors have stated that "One of the most abundant model protein for study of protein-ligand interactions is bovine serum albumin (BSA), because of its medical importance," but the reference cited (#4) is not related to the medical importance of BSA rather it is a simple study of the interaction of BSA with metal complexes.
  3. It is written in line No. 40 that BSA is whey protein. This sentence is wrong because BSA is a plasma protein, however, its traces could be found in the whey also but it is not the whey protein. Kindly correct this sentence or provide a relevant proof of that. 
  4. It is right that the amino acid sequence homology of BSA has 76 % of similarity to Human Serum Albumin (HSA) but it is completely wrong that its crystal structure is undefined until today. There are many crystal structures of BSA in the PDB database and one structure has already given in the manuscript in Fig. 9. 
  5. The writing of the introduction section is incoherent. The authors started with the importance of protein-ligand interactions and then came to the importance of BSA and then again switched to protein-ligand interaction. 
  6. No need to show the 3d-fluorescence spectrum (whose resolution is also not good) of pure BSA (Figure 5) because it has already been reported countless times, although a comparison between the spectra of pure and complexed BSA would be better.
  7. There are so many flaws in fluorescence measurements because the change in fluorescence around 4% (maximum around 6% in two cases) which includes from lowest to highest (because there is no consistency in fluorescence results) does not guarantee an interaction is taking place or not. That could have made sense if there was significant decrease or increases in the fluorescence followed by the reverse effect. The clear picture can easily be understood by simply seeing the fluorescence quenching/enhancement spectra which have not been shown by the authors in the manuscript. In my opinion, it is only an experimental error and there is no significant interaction between the ligands and proteins as far as the fluorescence spectroscopy is concerned because it depends only on the tryptophanyl environment. The other techniques may be more helpful for achieving this. It is recommended to study the interaction at higher concentrations of the ligands because it might be a weak interaction and the higher concentrations of the ligands may quench the tryptophan fluorescence significantly and may give the clear picture about the actual quenching/binding constants. ITC could also be an alternative of this. 
  8. What is the explanation of the negative quenching constants given in Table 4?
  9. The UV visible spectrum of the ligands must also be shown when dealing with steady-state fluorescence spectroscopy because the absorption of ligand at or near the excitation and emission wavelengths plays important role in fluorescence quenching that is called as inner filter effect which must be corrected before analyzing the fluorescence quenching data. 
  10. Line 245: tyrosine also participates in the fluorescence at 280 nm excitation. 
  11. There is no need to study the protein-ligand interaction at 225 nm excitation because excitation either at 280 nm or 295 nm could give the sufficient information. The excitation of protein at or near this wavelength has been criticized in a review (DOI: 10.1039/c6ra23426g) and the authors claimed that this fluorescence peak does not provide any information on backbone conformation, but simply reports on the local environment around the aromatic side chains, just as any traditional protein emission spectrum.
  12. Did authors see the fluorescence of the cyclitols only under experimental conditions? 
  13. Both FTIR and Raman techniques are excellent techniques for understanding protein interactions and conformational changes but ideally the contribution of the ligand intensity must be subtracted from the protein-ligand complex in order to get the clear picture of peak shifting and intensity change otherwise the ligand intensity will distort the results. There is no clarification whether the contribution of the ligands have been subtracted from the spectra of the protein-ligand complexes or not? 
  14. It is also recommended to confirm the binding sites of the studied cyclitols inside BSA using computational tool like molecular docking.
  15. Please provide the product numbers of the materials used particularly for BSA (fatted or defatted).
  16. Its better to write the molar concentrations of the stock solutions of the ligands.

Author Response

Reviewer #1

We are very grateful for your critical comments and thoughtful suggestions. Based on your suggestions, we have made a careful revision of the original manuscript. A revised manuscript has been submitted where the modified sections are provided in red color. We are very thankful for your contributions to improve our paper.

This paper discusses the interactions of four natural products with bovine serum albumin. They have used a number of methods/techniques to investigate the interaction. The topic of study is interesting and this could be a good paper but in my opinion there are so many flaws both in writing and analyses. The English of the manuscript also need through revision. Fluorescence studies have not been carried out accurately and need to be repeated or omitted from the paper. My comments to the authors are as follows:

  1. The importance of working with cyclitols should be elaborated in the introduction section. Why do the authors think that the study of these compounds with Bovine serum albumin would be of interest? Further, the description and importance of each cyclitol must be given in the introduction section. Overall introduction section should be rewritten.

Answer: Authors appreciate the Reviewer’s suggestion. The introduction part was modified to introduce more information about cyclitols themselves. Among others, the information about spectrum of biological activity of each individual cyclitol was presented. Moreover, the prospective utilization of cyclitols as nutraceuticals was defined.

  1. Authors have stated that "One of the most abundant model proteins for study of protein-ligand interactions is bovine serum albumin (BSA), because of its medical importance," but the reference cited (#4) is not related to the medical importance of BSA rather it is a simple study of the interaction of BSA with metal complexes.

Answer: Authors are thankful for this valuable comment. The introduction part was supplemented with additional information about BSA protein for the investigation in the biomedical field. The provided statements about BSA are supported by appropriate literature sources.

  1. It is written in line No. 40 that BSA is whey protein. This sentence is wrong because BSA is a plasma protein, however, its traces could be found in the whey also but it is not the whey protein. Kindly correct this sentence or provide a relevant proof of that.

Answer: Authors are grateful for this comment. The corresponding sentence has been rephrased, so authors believe that at the moment it presents the information about BSA in more accurate way.

  1. It is right that the amino acid sequence homology of BSA has 76 % of similarity to Human Serum Albumin (HSA) but it is completely wrong that its crystal structure is undefined until today. There are many crystal structures of BSA in the PDB database and one structure has already given in the manuscript in Fig. 9. 

Answer: Authors appreciate the Reviewer’s comment. The introduction part regarding BSA structure was rewritten to provide the information in more accurate way.

  1. The writing of the introduction section is incoherent. The authors started with the importance of protein-ligand interactions and then came to the importance of BSA and then again switched to protein-ligand interaction. 

Answer: Authors are grateful for this comment. Based on this comment the introduction part was reorganizes. Author believe that now it presents the information in more clear and consequent way, which will allow to familiarize the readers with the main idea of the performed investigations and manuscript in general.

  1. No need to show the 3d-fluorescence spectrum (whose resolution is also not good) of pure BSA (Figure 5) because it has already been reported countless times, although a comparison between the spectra of pure and complexed BSA would be better.

Answer: Authors appreciate the Reviewer’s comment. The additional investigations were performed to show the changes in the 3D-fluorescence of the synthesized cyclitol-BSA complexes in comparison to control. Briefly, the synthesis of cyclitol-BSA complexes were performed with utilization of high cyclitols concentrations (protein: cyclitol ratio 1:1000). The respective complexes were purified from unbonded cyclitols by utilization of several washing steps with deionized water and ultrafiltration. As a result, in the as synthesized complexes the fluorescence intensity from tryptophan residues ex./em. = 280/340 has decreased significantly (from nearly 33% for D-sorbitol and up to 60% for quinic acid). One of the reasons of such fluorescence quenching may be due to the loss of 3d-structure by BSA. However, the tryptophan band is sensitive to local environment and the destabilization of tertiary structure of the BSA usually leads to the peak maximum shifting, which was not observed in our study. Thus, the decrease in fluorescence intensity should be due to the cyclitols binding to BSA [22]. The appropriate information was included in the main body of the manuscript. The fluorescence spectra for blank and control solutions were also included in the supplementary files.

  1. Papadopoulou, A.; J Green, R.; A Frazier, R. Interaction of flavonoids with Bovine Serum Albumin: A Fluorscence Quenching Study. Journal of Argicultural and Food Chemistry 2005, 53, 58–63, doi.org/10.1021/jf048693g.

  1. There are so many flaws in fluorescence measurements because the change in fluorescence around 4% (maximum around 6% in two cases) which includes from lowest to highest (because there is no consistency in fluorescence results) does not guarantee an interaction is taking place or not. That could have made sense if there was significant decrease or increases in the fluorescence followed by the reverse effect. The clear picture can easily be understood by simply seeing the fluorescence quenching/enhancement spectra which have not been shown by the authors in the manuscript. In my opinion, it is only an experimental error and there is no significant interaction between the ligands and proteins as far as the fluorescence spectroscopy is concerned because it depends only on the tryptophanyl environment. The other techniques may be more helpful for achieving this. It is recommended to study the interaction at higher concentrations of the ligands because it might be a weak interaction and the higher concentrations of the ligands may quench the tryptophan fluorescence significantly and may give the clear picture about the actual quenching/binding constants. ITC could also be an alternative of this. 

Answer: Authors are thankful for this valuable comment. The comment given by the Reviewer is indeed critical. We have eliminated from manuscript previously shown data on fluorescence investigations based on Stern-Volmer model. Instead, according to Reviewer’s recommendation the simple changes in the fluorescence intensity after cyclitol binding was investigated and included to the manuscript (see the answer for comment Nr. 6). Moreover, we have succeeded to perform the ITC measurements for two cyclitols D-sorbitol and quinic acid (the members of linear and cyclic group respectively). Other representatives should interact with BSA in a similar way. The obtained results indicate that the cyclitols bind to BSA through weak interactions with low specificity. The hydrogen bonds and electrostatic interactions should be the main forces taking part in the binding process. The binding of the quinic acid did not cause the significant heat generation and in case of D-sorbitol it even requires the heat absorption to be bind to the protein. However, the changes in the entropy can be observed, which in consequence leads to the decrease in the free Gibbs energy of the system. The calculated values of changes in the free Gibbs energy are complementary to that one obtained from kinetic studies and molecular modeling. The binding of cyclitols to BSA should release the water molecules from the protein surface which increases the entropy of the system. In case of quinic acid, it possesses lower number of hydroxylic groups but one carboxylic group in comparison to D-sorbitol. Thus, the quinic acid binding should cause the desorption of lower amount of water molecules from protein surface, but occurrence of electrostatic interactions between cyclitol and BSA. Thus, it is reflected by respective thermodynamic parameters. Moreover, as it can be seen from kinetic studies the D-sorbitol and adonitol has a delay in the sorption in comparison to quinic and shikimic acid which, among others, may be connected to the heat effects of the reaction.

  1. What is the explanation of the negative quenching constants given in Table 4?

Answer: Thank You for this comment. The manuscript was modified significantly and the indicated Table and respective text was removed from the main discussion.

  1. The UV visible spectrum of the ligands must also be shown when dealing with steady-state fluorescence spectroscopy because the absorption of ligand at or near the excitation and emission wavelengths plays important role in fluorescence quenching that is called as inner filter effect which must be corrected before analyzing the fluorescence quenching data.

Answer: Authors are grateful for this comment. According to the Reviewer’s suggestions, the UV-Vis spectra of the ligands were recorded for the elimination of the possible influence of the Inner Filter Effect in the fluorescence study. The respective UV-Vis spectra are presented in the supplementary materials. Moreover, according to obtained data the methodology for the proper study and description of the fluorescence features of cyclitol-BSA interactions was adjusted.

  1. Line 245: tyrosine also participates in the fluorescence at 280 nm excitation. 

Answer: Thank You for this valuable comment. The corresponding sentence was modified. Thus, the authors believe that now it provides the information in a proper way.

  1. There is no need to study the protein-ligand interaction at 225 nm excitation because excitation either at 280 nm or 295 nm could give the sufficient information. The excitation of protein at or near this wavelength has been criticized in a review (DOI: 10.1039/c6ra23426g) and the authors claimed that this fluorescence peak does not provide any information on backbone conformation, but simply reports on the local environment around the aromatic side chains, just as any traditional protein emission spectrum.

Answer: Authors are thankful for this valuable comment. According to Reviewer’s suggestion, the manuscript was modified significantly and the band with excitation at 225 nm was not taken into account for the description of cyclitol-BSA interactions.

  1. Did authors see the fluorescence of the cyclitols only under experimental conditions? 

Answer: Thank You for your question. We have performed the measurements of the cyclitols fluorescence under the experimental conditions. The respective 3d-fluorescence spectra are presented in the supplementary materials. The results indicates that fluorescence of cyclitols solutions do not differ significantly from fluorescence of pure water. Only on the spectrum of quinic acid low intensity peak at λex./em. of 250/445 nm can be distinguished, which rather have no influence on the obtained results of cyclitol-BSA interactions.

  1. Both FTIR and Raman techniques are excellent techniques for understanding protein interactions and conformational changes but ideally the contribution of the ligand intensity must be subtracted from the protein-ligand complex in order to get the clear picture of peak shifting and intensity change otherwise the ligand intensity will distort the results. There is no clarification whether the contribution of the ligands have been subtracted from the spectra of the protein-ligand complexes or not? 

Answer: Authors appreciate the Reviewer’s critical comment. According to Reviewer’s comment we have change the way in which the results are presented. However, it should be noted that FTIR and Raman are techniques that has quantitative character, e. g. the intensity of the signals is proportional to the number of particular functional groups. Our investigations were performed in the way that does not allow to evaluate the impact of cyclitols on cyclitol-BSA spectra quantitatively, as the amount of applied cyclitols on CaF2 lenses and glass was much higher than it remains in the complex. Thus, the substruction of cyclitol spectrum from corresponding cyclitol-BSA spectrum will be incorrect approach. Nevertheless, now on the figures for each cyclitol three spectra are presented: BSA, cyclitol-BSA and cyclitol itself. As can be seen from the figure 6, on the FTIR spectra the bands from free BSA and cyclitols appears in different banding regions. In case of cyclitol-BSA spectrum the higher signals from BSA can be observed (e.g. amide I, amide II, etc.), but with changes induced by cyclitol binding. In case of Raman the situation mostly similar to FTIR results.

  1. It is also recommended to confirm the binding sites of the studied cyclitols inside BSA using computational tool like molecular docking.

Answer: Authors are grateful for this comment. According to the Reviewer’s comment the appropriate investigation has been performed. The Autodock software was chosen to perform the calculations as it is free for the consumer, simple for usage and not time consuming. The results of the generated structures are presented on the figure 8. Moreover, the binding energies was able to derive from the calculations which are complementary to the energies obtained by ITC measurements and kinetic study.

  1. Please provide the product numbers of the materials used particularly for BSA (fatted or defatted).

Answer: Thank You for the question. The BSA was supplied by Sigma-Aldrich company. The product number was A7638. The supplier did not specify if the protein was defatted. However, according to the presented information on the supplier’s web page, the protein purification methods, among others, include ethanol fractionation and charcoal filtration which should remove such impurities as fats.

  1. Its better to write the molar concentrations of the stock solutions of the ligands.

Answer: Thank You for this comment. Appropriate changes in the Materials and methods section were introduced.

Reviewer 2 Report

The manuscript is well written. Experiments were planned correctly with the results that has been presented clearly and seems to be conclusive. I have some minor objections/questions that should be take into consideration.

  1. In line 177 (description for table 2) Authors written an abbreviation “[2M-H]+” but in the table 2 is lack of it. Is it correct?
  2. I strongly recommend improve the quality of figure 4. Description of “X” and “Y” axis are not good readable. Also at figure 7 font at “X” and “Y” axis should be larger.
  3. One from techniques used by Authors is fluorescence spectroscopy. I have some questions: (a) Did Authors verified if cyclitols did not absorb the light at the wavelengths used for excitation and emission of fluorescence? If they absorb then the correction of fluorescence values for the inner filter effect should be applied, (b) For the excitation wavelength 280nm not only tryptophan’s but also tyrosine’s (in BSA is 20 tyrosine’s residues) fluorescence is excited. Thus fluorescence is gain both from Trp and Tyr residues. Did Authors take into consideration that maybe non-linear Stern-Volmer plots can be also the result of fluorescence obtained for more than 1-2 fluorophores? Maybe better is use excitation wavelength 295 nm. Then the fluorescence will be measured just from Trp because at 295nm the absorption coefficient for tyrosine is very low and the Tyr fluorescence is eliminate.
  4. In line 45 should be written “domains (I, II, III)” instead of “domains (I, II, II)”
  5. In lines: 54, 331, 669 are repetitive words. Please delete one from it.
  6. In line 446 should be written “can” instead of “cam”
  7. In line 517 should be written “bonding” instead of “bonging”

Author Response

Reviewer #2

We are very grateful for your critical comments and thoughtful suggestions. Based on your suggestions, we have made a careful revision of the original manuscript. A revised manuscript has been submitted where the modified sections are provided in red color. We are very thankful for your contributions to improve our paper.

  1. In line 177 (description for table 2) Authors written an abbreviation “[2M-H]+” but in the table 2 is lack of it. Is it correct?

Answer: Authors are thankful for your valuable comment. The Reviewer’s comment indeed critical. During the cyclitol identification we wanted to check if these molecules will form different types of adducts by ionization. No dimer for adonitol was observed in the positive mode of ionization. The abbreviation “[2M-H]+” was removed from text as it was surplus.

  1. I strongly recommend improve the quality of figure 4. Description of “X” and “Y” axis are not good readable. Also at figure 7 font at “X” and “Y” axis should be larger.

Answer: Authors appreciate the Reviewer’s comment. We have improved the quality of figure 4 in the proper way, according to your recommendation. The data from spectra at figure 7 (figure 6 in new version) were processed in the OriginPro 2016 software, we hope that parameters of prepared figures became much better.

  1. One from techniques used by Authors is fluorescence spectroscopy. I have some questions: (a) Did Authors verified if cyclitols did not absorb the light at the wavelengths used for excitation and emission of fluorescence? If they absorb then the correction of fluorescence values for the inner filter effect should be applied, (b) For the excitation wavelength 280nm not only tryptophan’s but also tyrosine’s (in BSA is 20 tyrosine’s residues) fluorescence is excited. Thus fluorescence is gain both from Trp and Tyr residues. Did Authors take into consideration that maybe non-linear Stern-Volmer plots can be also the result of fluorescence obtained for more than 1-2 fluorophores? Maybe better is use excitation wavelength 295 nm. Then the fluorescence will be measured just from Trp because at 295nm the absorption coefficient for tyrosine is very low and the Tyr fluorescence is eliminate.

Answer: (a) We are grateful for this valuable comment. According to Reviewer’s comment we perform additional measurements comprised of UV-Vis absorption and fluorescence study of cyclitols. Results displayed that maximum of absorption for shikimic and D-(-)-quinic acid was close to 225 nm. Instead, none of analysed cyclitols revealed the absorbance at 280 nm. Moreover, the 3D spectra showed no fluorescence for cyclitol molecules at 340 nm. Thus, according to the obtained data the fluorescence changes for cyclitol-BSA complexes was monitored on the band λex./em. = 280/340 nm

(b) Authors appreciate the Reviewer’s comment. We took into account the contribution of tyrosine for BSA fluorescence. The authors decided to omit the Stern-Volmer plots because of many flaws. In order to evaluate the influence of ligand on the protein conformation, the authors considered to measure 3D fluorescence spectra for pure BSA and BSA-cyclitol complexes and compare them between each other. Two peaks were observed for native BSA: peak 1 (λEx = 225 nm and λEm = 340 nm) and peak 2 (λEx = 280 nm and λEm = 340 nm). As Inner-filter effect is possible for peak 1 (due to absorption of shikimic and D-(-)-quinic acid at 225 nm), thus the changes of fluorescence intensity were monitored for peak 2. The results showed that cyclitols induced the quenching of BSA fluorescence which indicates the protein-ligand interactions in the close proximity.

  1. In line 45 should be written “domains (I, II, III)” instead of “domains (I, II, II)”

Answer: Thank You for your remark, the sentence was corrected.

  1. In lines: 54, 331, 669 are repetitive words. Please delete one from it.

Answer: Thank You very much for your remark, the doubled word was removed from the text.

  1. In line 446 should be written “can” instead of “cam”

Answer:  Thank You for your remark, the word was corrected.

  1. In line 517 should be written “bonding” instead of “bonging”

Answer: Thank You for your remark, the word was corrected.

Reviewer 3 Report

The subject of the research is the analysis of the interaction between bovine serum albumin and four compounds from the cyclitol group. The research was planned in a way that distinguishes it from other papers on this subject. Unfortunately, there are a number of errors and inaccuracies in the manuscript.

Line 40 - BSA is not a whey protein.

Line 44 - BSA structure is quite well described. Probably a translation error.

In Chapter 2.2, the Authors provided an extensive interpretation of the BSA-cyclitol interaction based on the measurement of the pH of the albumin-cyclitol mixture. Adonitol and D-sorbitol have a similar structure, and also pH (~ 5.5) of the standard solution. Similarly, shikimic acid and D-quinic acid have a similar pH (~ 4.0) and structure. The addition of the individual components resulted in a similar curve shape of the kinetic model shown in Fig. 2 (A similar to C, and B to D). Ligands were added in a 10:1 molar ratio with respect to BSA. Did the Authors check whether the addition of ligands at this concentration did not exceed the BSA buffer capacity, and did not lower the pH of the solutions by the mere addition of cyclitol? Will the addition of an agent with a different structure but similar pH and ionic strength result in a different shape of the curve of the kinetic model?

Lines 100-103 - "Autocatalysis of water molecules promotes the hydrogen ions transfer into solution as well as increase the availability of hydroxyl groups on the protein surface inducing the hydrolysis of deprotonated carboxyl groups"

The above sentence is incomprehensible to me. Perhaps because catalysis is not my area of expertise. The literature given as reference [17] relates to a model of CO2 reduction to methanol using a copper catalyst.

Lines 194-196 - "It may be a result of their cyclic structure and steric interactions as well as electrostatic interactions between deprotonated carboxylic group and charged groups of BSA "

pH of the mobile phase was probably below 3.0 (0.1% formic acid is about 2.7), so I don't think there will be many deprotonated carboxylic groups in cyclitols or albumin. Maybe except quinic acid (which has a longer retention time, as seen on Fig. 4D)

Chapter 2.4 - The Authors, again for fluorescence spectroscopy, conducted a very extensive interpretation of the obtained results. However, very small changes of the F0/F parameter are puzzling. In my opinion they are very close to the measurement error. Did the Authors manage to repeat the analysis, obtaining identical results? In addition, a number of results for the F0/F parameter are below the value of 1 (and therefore the Ksv values in Table 4 are negative) - which clearly indicates an enhancement of the fluorescence effect. This conclusion is obvious especially for quinic acid. Did the Authors check whether the ligands themselves do not fluoresce under these conditions? In my opinion, based on the results presented in the manuscript, it is impossible to draw conclusions regarding the interaction of cyclitols with albumin.

Chapter 2.5 – Fig. 7A and 7B - BSA-cyclitol spectra de facto show the spectra of individual cyclitols, completely obscuring the albumin bands. This rules out the possibility of any interpretation of the spectra in terms of albumin stability, its second-order structure and conformational changes, as assumed at the beginning of the Chapter (lines 360,361). Perhaps, if the Authors took into account the spectra of cyclitols by subtracting the spectra of pure substances from the spectrum of the BSA-cyclitol complex, some changes in the BSA spectrum could be observed.

Chapter 2.6 - I have a similar objections to the interpretation of the results obtained by Raman spectroscopy. Without comparison with the Raman spectra of individual cyclitols, a detailed interpretation of the changes in the BSA spectra seems impossible. The Authors also did not explain what happened to the C-H stretching oscillations visible on the spectrum of pure albumin and not visible on the spectra of the complexes.

Additionally, Raman spectra are signed with the number 7 identically to FTIR spectra.

Figure 9 shows the interpretation of the interaction of albumin with individual ligands. It is a pity that the authors did not try to make a computer model of the complex, e.g. using the free Autodock software. The BSA model is available so this task is not very time consuming and would be a valuable addition.

In summary, the manuscript was written with a lot of work, but in my opinion it still requires a number of corrections. Especially the methodological errors that I showed above must be corrected. The translation into English also needs to be improved, as there are many linguistic and grammatical errors in the text.

Author Response

Reviewer #3

We are very grateful for your critical comments and thoughtful suggestions. Based on your suggestions, we have made a careful revision of the original manuscript. A revised manuscript has been submitted where the modified sections are provided in red color. We are very thankful for your contributions to improve our paper.

  1. Line 40 - BSA is not a whey protein.

Answer: Thank You very much for your comment. Obviously, BSA belongs to serum proteins. The sentence was corrected.

  1. Line 44 - BSA structure is quite well described. Probably a translation error.

Answer: Authors appreciate this valuable comment. Of course, BSA structure is clearly described in PDB database. The error was eliminated.

  1. In Chapter 2.2, the Authors provided an extensive interpretation of the BSA-cyclitol interaction based on the measurement of the pH of the albumin-cyclitol mixture. Adonitol and D-sorbitol have a similar structure, and also pH (~ 5.5) of the standard solution. Similarly, shikimic acid and D-quinic acid have a similar pH (~ 4.0) and structure. The addition of the individual components resulted in a similar curve shape of the kinetic model shown in Fig. 2 (A similar to C, and B to D). Ligands were added in a 10:1 molar ratio with respect to BSA. Did the Authors check whether the addition of ligands at this concentration did not exceed the BSA buffer capacity, and did not lower the pH of the solutions by the mere addition of cyclitol? Will the addition of an agent with a different structure but similar pH and ionic strength result in a different shape of the curve of the kinetic model?

Answer: Authors are thankful for your valuable comment. It is well-known that protein solutions itself can serve as weak buffer system. During our investigation the pH was measured for the pure protein solution. The measured pH of BSA control did not change during the whole time of experiment, thus protein buffer capacity was not exceeded during the experiment by means of CO2 dissolution or other changes caused by reaction conditions. Instead, the changes in the pH after cyclitols addition is significant. For D-sorbitol and adonitol the same sharp shape of the kinetic modelling curve was observed. It may be related to high similarity of their structures. The kinetic curve shape for shikimic acid and D-(-)-quinic differed from D-sorbitol and adonitol which indicates the different character of interactions of linear- and cyclic-structure molecules with BSA. Influence of other agents with a different structure but similar pH and ionic strength on the kinetic curve shape was not performed. 

  1. Lines 100-103 - "Autocatalysis of water molecules promotes the hydrogen ions transfer into solution as well as increase the availability of hydroxyl groups on the protein surface inducing the hydrolysis of deprotonated carboxyl groups"

The above sentence is incomprehensible to me. Perhaps because catalysis is not my area of expertise. The literature given as reference [17] relates to a model of CO2 reduction to methanol using a copper catalyst.

Answer: Authors are thankful for this valuable comment. The authors have utilized the wrong conception for mechanisms description. The corresponding part was modified, i.e. the sentence was corrected to: “Autoionization of water molecules promotes the increase of hydroxyl groups availability to protein surface while hydronium ions participate in neutralization of deprotonated carboxyl groups.”

  1. Lines 194-196 - "It may be a result of their cyclic structure and steric interactions as well as electrostatic interactions between deprotonated carboxylic group and charged groups of BSA "

pH of the mobile phase was probably below 3.0 (0.1% formic acid is about 2.7), so I don't think there will be many deprotonated carboxylic groups in cyclitols or albumin. Maybe except quinic acid (which has a longer retention time, as seen on Fig. 4D).

Answer: Authors appreciate the Reviewer’s comment. The additional measurements were performed. We have checked that pH of mobile phase was 2.61, thus, carboxylic groups of shikimic (pKa = 4.1) and D-(-)-quinic acid (pKa = 3.5) tend to be in protonated form. The corresponding corrections were introduced to the main text, i. e. that contribution of electrostatic interactions between charged groups of BSA and these cyclitols were negligible.

  1. Chapter 2.4 - The Authors, again for fluorescence spectroscopy, conducted a very extensive interpretation of the obtained results. However, very small changes of the F0/F parameter are puzzling. In my opinion they are very close to the measurement error. Did the Authors manage to repeat the analysis, obtaining identical results? In addition, a number of results for the F0/F parameter are below the value of 1 (and therefore the Ksv values in Table 4 are negative) - which clearly indicates an enhancement of the fluorescence effect. This conclusion is obvious especially for quinic acid. Did the Authors check whether the ligands themselves do not fluoresce under these conditions? In my opinion, based on the results presented in the manuscript, it is impossible to draw conclusions regarding the interaction of cyclitols with albumin.

Answer: Authors are thankful for this valuable comment. The comment given by the Reviewer is indeed critical. We have eliminated from manuscript previously shown data on fluorescence investigations based on Stern-Volmer model. Instead, the influence of ligand on the protein conformation was determined by comparison of 3D fluorescence spectra for pure BSA and BSA-cyclitol complexes. Cyclitol molecules revealed not strong fluorescence, which may partly origin from negative control (cuvette with water). Particularly, the UV-VIS spectra of adonitiol, shikimic acid and D-(-)-quinic acid displayed the emission peak at 540, 290, 445 nm, respectively, which wasn’t covered with characteristic protein fluorescence at 340 nm.

  1. Chapter 2.5 – Fig. 7A and 7B - BSA-cyclitol spectra de facto show the spectra of individual cyclitols, completely obscuring the albumin bands. This rules out the possibility of any interpretation of the spectra in terms of albumin stability, its second-order structure and conformational changes, as assumed at the beginning of the Chapter (lines 360,361). Perhaps, if the Authors took into account the spectra of cyclitols by subtracting the spectra of pure substances from the spectrum of the BSA-cyclitol complex, some changes in the BSA spectrum could be observed.

Answer: We are grateful for this valuable remark. The Fig. 7A as well as 7B were divided into four areas; each of them presents the spectra of each individual cyclitol, complex and native BSA. Some band shifting occurred after data normalization and baseline correction by OriginPro2016 software. The changes in BSA-cyclitol spectra were analysed by subtraction of the individual cyclitol. Moreover, the peak intensity wasn’t taken into account anymore, because the experiment wasn’t performed quantitatively.

  1. Chapter 2.6 - I have a similar objections to the interpretation of the results obtained by Raman spectroscopy. Without comparison with the Raman spectra of individual cyclitols, a detailed interpretation of the changes in the BSA spectra seems impossible. The Authors also did not explain what happened to the C-H stretching oscillations visible on the spectrum of pure albumin and not visible on the spectra of the complexes.

Answer: Authors appreciate the Reviewer’s comment. According to Reviewer’s comment we have change the way in which the results are presented. However, it should be noted that FTIR and Raman are techniques that has quantitative character, e. g. the intensity of the signals is proportional to the number of particular functional groups. Our investigations were performed in the way that does not allow to evaluate the impact of cyclitols on cyclitol-BSA spectra quantitatively, as the amount of applied cyclitols on CaF2 lenses and glass was much higher than it remains in the complex. Thus, the substruction of cyclitol spectrum from corresponding cyclitol-BSA spectrum will be incorrect approach. Nevertheless, now on the figures for each cyclitol three spectra are presented: BSA, cyclitol-BSA and cyclitol itself. As can be seen from the figure 6, on the FTIR spectra the bands from free BSA and cyclitols appears in different banding regions. In case of cyclitol-BSA spectrum the higher signals from BSA can be observed (e.g. amide I, amide II, etc.), but with changes induced by cyclitol binding. In case of Raman the situation mostly similar to FTIR results.

  1. Additionally, Raman spectra are signed with the number 7 identically to FTIR spectra.

Answer: Thank you for your remark, the figure number was corrected.

  1. Figure 9 shows the interpretation of the interaction of albumin with individual ligands. It is a pity that the authors did not try to make a computer model of the complex, e.g. using the free Autodock software. The BSA model is available so this task is not very time consuming and would be a valuable addition.

Answer: Authors appreciate the Reviewer’s comment. According to the Reviewer’s suggestion the authors provided the study of protein-cyclitol interactions by molecular docking. Autodock software was easy and comfortable in operation. The obtained results are quite consistent with experimental data.

  1. The translation into English also needs to be improved, as there are many linguistic and grammatical errors in the text.

Answer: Thank you very much for your comment. The manuscript was subjected to an editing procedure to improve the quality of the text in accordance to standard English.

Round 2

Reviewer 1 Report

Title: The study of protein-cyclitol interactions Authors: Tetiana Dyrda-Terniuk , Magdalena Buszewska-Forajta , PaweÅ‚ Pomastowski * , BogusÅ‚aw Buszewski   After careful evaluation of the revision of the manuscript, I found that authors have performed an extensive revision and now the manuscript looks fine. However, I would like to suggest a minor change; please put the ITC section before the molecular docking so that the experimental portion comes together followed by the computational part. Rest of the revision is well done and I recommend the publication of the paper after performing the minor change described above. 

Author Response

Reviewer #1

We are very grateful for Reviewer’s comments and thoughtful suggestions. A revised manuscript has been submitted where the modified sections are provided in red color. We are very thankful for your contributions to improve our paper.

After careful evaluation of the revision of the manuscript, I found that authors have performed an extensive revision and now the manuscript looks fine. However, I would like to suggest a minor change; please put the ITC section before the molecular docking so that the experimental portion comes together followed by the computational part. Rest of the revision is well done and I recommend the publication of the paper after performing the minor change described above. 

Answer: Authors appreciate the Reviewer’s suggestion. ITC and molecular docking sections were exchanged.

Reviewer 3 Report

The second version of the mancript is significantly different from the original version. The Authors added new analytical methods (ITC and MD) and significantly changed the methodology and interpretation of some of the previously presented methods (FTIR, Raman and Fluorescence). The introduction has also been changed. The manuscript is significantly improved in terms of language and grammar. With such big changes, a few errors could not be avoided.

Lines 61,62 - "1,3,4,5-tetrahyroxycyclohexane-L-carboxylic acid" 
            missig letter d - "tetrahydroxy"

Line 100 - "...approach in drug designing and can to discover new drug candidates." 
            should be "...approach in drug design and can lead to discovery of new drug candidates." 

Lines 118-119 - "The bands on obtained electropherogram (Fig. 1) are placed in the range at about 65 kDa and 130 kDa"
            should be "The bands on obtained electropherogram (Fig. 1) are placed (or "are visible") at about 65 kDa and 130 kDa"

I think that the detailed list of wavenumbers in the captions of Figures 6 and 7 could be transferred to the table in the supplement.

Lines 775-775 - double "were"

Line 778 - "...and much effective quenching of protein..." 
            should be "...more effective..."

Line 788 - "For the establishment a more detailed mechanism..."
            should be "For the establishment of a more detailed mechanism..."

The above remarks are not of a substantive nature. After minor errors have been corrected, the manuscript will meet the requirements of a scientific publication.

Author Response

Reviewer #3

We are very grateful for Reviewer’s comments and thoughtful suggestions. A revised manuscript has been submitted where the modified sections are provided in red color. We are very thankful for your contributions to improve our paper.

  1. Lines 61,62 - "1,3,4,5-tetrahyroxycyclohexane-L-carboxylic acid" missig letter d - "tetrahydroxy".

Answer: Authors appreciate the Reviewer’s remark, the word was corrected.

  1. Line 100 - "...approach in drug designing and can to discover new drug candidates."  should be "...approach in drug design and can lead to discovery of new drug candidates." 

Answer: Authors are grateful for your remark, the sentence was corrected.

  1. Lines 118-119 - "The bands on obtained electropherogram (Fig. 1) are placed in the range at about 65 kDa and 130 kDa" should be "The bands on obtained electropherogram (Fig. 1) are placed (or "are visible") at about 65 kDa and 130 kDa".

Answer: Thank You very much for your remark, the sentence was corrected.

  1. I think that the detailed list of wavenumbers in the captions of Figures 6 and 7 could be transferred to the table in the supplement.

Answer: Authors appreciate the Reviewer’s suggestion. Detailed list of captioned wavenumbers of Fig. 6 and Fig. 7 are shown in the Supplementary at S.4 and S.5

  1. Lines 775-775 - double "were".

Answer: Thank You very much for your remark, the doubled word was removed from the text.

  1. Line 778 - "...and much effective quenching of protein..."  should be "...more effective...".

Answer: Authors are grateful for your remark, the phrase was corrected.

  1. Line 788 - "For the establishment a more detailed mechanism..." should be "For the establishment of a more detailed mechanism..."

Answer: Authors appreciate the Reviewer’s remark, the sentence was corrected.
